

# Employing SAE-GRU deep learning for scalable botnet detection in smart city infrastructure

Usman Tariq* and Tariq Ahamed Ahanger*

Prince Sattam Bin Abdulaziz University, Al-Kharj, Al-Riyadh, Saudi Arabia
* These authors contributed equally to this work.

## ABSTRACT

The proliferation of Internet of Things (IoT) devices in smart cities has revolutionized urban infrastructure while escalating the risk of botnet attacks that threaten essential services and public safety. This research addresses the critical need for intrusion detection and mitigation systems by introducing a novel hybrid deep learning model, Stacked Autoencoder–Gated Recurrent Unit (SAE-GRU), specifically designed for IoT networks in smart cities. The study targets the dual challenges of processing high-dimensional data and recognizing temporal patterns to identify and mitigate botnet activities in real time. The methodology integrates Stacked Autoencoders for reducing dimensionality and gated recurrent units for analyzing sequential data to ensure both accuracy and efficiency. An emulated smart city environment with diverse IoT devices and communication protocols provided a realistic testbed for evaluating the model. Results demonstrate significant improvements in detection performance with an average accuracy of 98.65 percent and consistently high precision and recall values. These findings enhance the understanding of IoT security by offering a scalable and resource-efficient solution for botnet detection. The functional investigation establishes a foundation for future research into adaptive security mechanisms that address emerging threats and highlights the practical potential of advanced deep learning techniques in safeguarding next-generation smart city ecosystems.

# INTRODUCTION

The growth of Internet of Things (IoT) devices in urban environments has significantly enabled advancements in urban management and an improved quality of life for residents. Yet, it has also introduced substantial challenges in ensuring security and privacy (*Hazman et al., 2024*). As smart cities evolve, the centrality of IoT devices necessitates robust systems capable of managing and safeguarding interconnected technologies. With IoT applications extending across traffic control, public safety, and environmental monitoring, the security of these devices becomes vital for the effective operation of modern urban systems (*Bhardwaj et al., 2024*).

Corresponding author
Usman Tariq, u.tariq@psau.edu.sa

The increasing integration of IoT devices has been accompanied by a parallel rise in security threats, particularly from IoT botnets that exploit vulnerabilities to carry out malicious activities (*Krishnan & Shrinath, 2024*). Recent high-profile incidents highlight the need for urgent enhancements to security measures. Botnets are capable of taking over large numbers of IoT devices, turning them into networks of remotely controlled bots that can initiate large-scale attacks, disrupting city operations and compromising sensitive data.

These escalating threats have made the need for effective intrusion detection systems (IDS) apparent (*Indra et al., 2024*). IDS are critical in detecting and neutralizing threats originating from IoT botnets, thereby preserving the integrity and reliability of smart city infrastructures. Without such robust IDS, smart city initiatives face potential disruptions that can undermine public trust and jeopardize essential services.

Traditional IDS, despite their importance, show significant limitations (*i.e.*, includes but not limited to: high false positive rates; poor scalability; incompatibility; limited real-time analysis, inflexibility; lack of contextual awareness; protocol incompatibility, reliance on signature-based detection, *etc.*,) when used in IoT environments. The heterogeneous and dynamic nature of interconnected sensing devices, coupled with scalability and resource limitations, presents challenges that conventional IDS solutions cannot adequately address (*Li et al., 2023*). Thus, techniques that once sufficed for simpler networks are no longer effective in tackling the diverse security requirements (such as: Real-time threat detection; Scalability for large-scale networks; low latency response; adaptability to diverse protocols; data integrity assurance; *etc.*,) presented by complex networks.

To address these limitations, deep learning (DL) technologies offer a promising solution to the challenges present in IoT security (*Zakariyya, Kalutarage & Al-Kadri, 2023*). These advanced methods can detect complex, subtle patterns that traditional approaches often miss which makes them well-suited for the intricacies of smart city networks. Deep learning models have demonstrated their capacity to enhance detection capabilities and adapt more effectively to evolving threats. Nonetheless, integrating these models into IoT intrusion detection poses its own challenges. The need for real-time processing and the diverse data types found in networks demand highly efficient and flexible models that operate effectively even with the typical limitations of devices, such as restricted processing power and limited energy resources.

Even with the progress made with DL-based IDS, significant gaps remain (*Shahin et al., 2024*). Many current systems fail to meet real-time processing needs or adequately manage the range of node-specific threats. These deficiencies underscore the necessity for more advanced models capable of addressing the unique challenges of IoT security. Motivated by these limitations, this research aims to develop and assess a novel deep learning-powered IDS specifically designed for IoT environments in smart cities. This study advances beyond the existing state of the art by addressing these nuanced needs through a new architectural approach.

## Significant contributions

The primary objective of this research is to push the boundaries of IoT security through the introduction of a novel deep learning architecture. The proposed Stacked Autoencoder–

[1] This (SAE-GRU) hybrid model employs stacked autoencoders to extract meaningful features by reducing redundant information before passing the refined data to gated recurrent units that capture sequential dependencies essential for intrusion detection in IoT networks.

[2] Stacked autoencoders performed dimensionality reduction by learning compressed representations of input data through an unsupervised learning process. This approach retained significant features while eliminating irrelevant variations which enhances the efficiency of subsequent classification tasks.

Gated Recurrent Unit (SAE-GRU[1]) model aims to enhance both the accuracy and efficiency of intrusion detection within networks. This article details the design, implementation, and deployment of the proposed model in a smart city context to furnish a comprehensive evaluation of its performance across several metrics to validate its effectiveness. Hereby, the main contributions of proposed research are listed below:

a. Introduced a hybrid deep learning model, SAE-GRU, that combines Stacked Autoencoders for dimensionality reduction[2] and gated recurrent units for temporal pattern recognition to detect botnet activities in smart city IoT networks.

b. Implemented model pruning and weight quantization techniques to reduce computational complexity and improve the efficiency of the intrusion detection system without compromising accuracy.

c. Developed an emulation environment replicating a smart city network using diverse IoT devices and communication protocols, which enabled comprehensive testing of the proposed system.

d. Integrated sparse matrix multiplication and batch processing to optimize the inference process to minimize computational overhead and guarantee real-time detection capabilities.

e. Applied truncated backpropagation through time in the GRU to manage sequential data processing effectively to effectively reduce latency in real-time applications.

f. Conducted rigorous performance evaluations using k-fold cross-validation and various datasets to demonstrate superior accuracy, precision, recall, and AUC values compared to traditional models.

g. Employed feature importance analysis techniques, including SHAP values to identify key features like packet size and traffic volume to contribute significantly to detection accuracy.

h. Demonstrated the model's scalability and robustness in handling high data volumes and diverse network conditions to make it suitable for large-scale smart city implementations.

i. Highlighted the model's resilience against zero-day attacks and its ability to adapt to evolving botnet strategies through advanced deep learning techniques.

j. Provided practical insights for future research by suggesting methods to enhance computational efficiency and extend the system's capability to handle more complex IoT network environments.

The remainder of this article is organized to provide a detailed exploration of each component of the proposed IDS. Following this introduction, the article presents a review of related works to set the context for this study, followed by an explanation of fundamental concepts essential for understanding IoT botnet detection. The subsequent sections describe the proposed methodology, emulation setup, results, and an in-depth analysis of the key features identified during the study. The article concludes with a

summary of findings and discusses their implications for future research and practical applications.

## RELATED WORKS

An IoT botnet within smart city network represents a conglomeration of compromised IoT devices integrated into urban infrastructures, orchestrated by malicious actors to conduct synchronized cyber-attacks that jeopardize the confidentiality, integrity, and availability of critical municipal services (*Alshahrani, 2023*). These botnets capitalize on intrinsic vulnerabilities such as insecure firmware, inadequate authentication protocols, and unencrypted communication channels like Message Queuing Telemetry Transport (MQTT) & Constrained Application Protocol (CoAP) which infiltrates in devices from environmental sensors to satellite communication modules. Novel and ultra-advanced botnet attacks employ sophisticated methodologies, including decentralized peer-to-peer command and control architectures, polymorphic malware that adapts its code to evade detection, and the use of machine learning algorithms to tailor attack vectors dynamically (*Kornyo et al., 2023*). Attack strategies may encompass cross-layer exploits targeting physical, network, and application layers simultaneously or manipulate satellite communication pathways to disseminate malicious payloads across a vast array of IoT devices, thereby amplifying the attack's scale and complexity. The availability of extensive botnet datasets (*Kalakoti, Bahsi & Nõmm, 2024*), such as MedBIoT and IoT-23, delivers researchers with critical empirical data to analyze network traffic anomalies, model malware behavior, and develop advanced intrusion detection systems using machine learning techniques, ultimately enhancing the defensive posture against evolving botnet threats.

*Hazman et al. (2024)* proposed an IDS that combines deep learning with feature engineering to improve threat detection in IoT-based smart cities. The core of the framework is a long short-term memory (LSTM) model enhanced by autoencoders (AEs), genetic algorithms (GAs), and information gain (IG) for dimensionality reduction and input optimization. This integration improved classification efficiency in handling high-dimensional, imbalanced IoT data. The IDS was validated on datasets such as BoT-IoT, Edge-IIoT, and NSL-KDD, which demonstrated high accuracy, precision, and recall with notably low false positive and false negative rates. A key strength of this research was the use of Tensor Processing Units (TPUs), which significantly reduced training and classification time in real-time deployment. Furthermore, the feature engineering pipeline enhanced data quality and detection performance, especially against complex threats like Distributed Denial of Service (DDoS) and reconnaissance attacks. The limitations of model included potential overfitting due to reliance on labeled data and reduced generalizability across varied IoT infrastructures. Despite these, the study marked a meaningful step toward improving IDS performance in smart city environments.

*Almasri & Alajlan (2023)* introduced a two-phase deep learning model for detecting and isolating cyber-attacks in IoT-based smart city systems. The first phase utilized a cascaded adaptive neuro-fuzzy inference system (CANFIS) to identify malicious traffic, detect compromised devices, and isolate them. In the second phase, modified deep reinforcement

learning (MDRL) blocked communication channels of the infected devices to prevent further threats. The model achieved 98.7% detection accuracy by outperforming LSTM, support vector machines (SVM), and standard deep reinforcement learning (DRL) models across precision, recall, and F1-score. Its ability to significantly reduce detection time supported real-time threat mitigation. Validation on IoT network intrusion and ISCX 2012 datasets confirmed its generalizability. CANFIS's sequential feature selection improved precision detection, while MDRL contributed to adaptive response strategies. A key limitation was computational demand which made its deployment on resource-constrained edge devices challenging. Despite this, the model's high accuracy and low false positive rate make it a strong candidate for smart city security, with further potential in industrial and healthcare IoT through efficiency optimization.

*Taher et al. (2023)* proposed a machine learning framework to improve botnet detection in Industrial IoT systems. The approach integrates a hybrid feature selection method, FGOA-kNN, which combines Fisher-score, Grasshopper Optimization algorithm, and k-nearest neighbor to identify relevant features and eliminate redundancy with a focus to improve accuracy and computational efficiency. The model also incorporated an optimized neural network, IHHO-NN that was fine-tuned using an enhanced Harris Hawks Optimization algorithm to classify multiclass botnet attacks with high precision. Validated on the N-BaIoT dataset, the model outperformed traditional classifiers in accuracy, recall, and precision for both known and novel attacks. Its combination of unsupervised clustering with supervised learning strengthened robustness in handling high-dimensional, complex IIoT data. Improvements in convergence *via* chaotic maps and Random Opposition-Based Learning allowed real-time use in constrained settings. During our investigation, we experienced a few limitations, such as dependence on the N-BaIoT dataset, which raised concerns about generalizability across varied IoT systems and evolving threats.

*Manickam et al. (2023)* presented a novel integration of Billiard Based Optimization (BBO) and deep learning for anomaly detection in IoT-enabled smart cities while focusing/emphasizing on sustainable applications. The model employed Binary Pigeon Optimization (BPEO) for feature selection, Elman recurrent neural network (ERNN) for anomaly detection, and BBO for hyperparameter tuning to achieve high accuracy in anomaly classification. The main contribution lies in its ability to handle IoT data's complexity and identify patterns in resource-constrained environments with an aim to enhance both detection accuracy and computational efficiency. The model achieved outstanding performance with an accuracy rate exceeding 99% on benchmark datasets. This result highlights its ability to capture temporal and spatial patterns while reducing false positives. Nonetheless, the reliance on centralized data processing presented a limitation, as it may lead to latency and increased vulnerability to network congestion in large-scale IoT systems. Furthermore, the model's dependence on specific datasets raised concerns about its capacity to generalize effectively across diverse and evolving IoT environments.

*Hazman et al. (2023)* demonstrated significant advancements in IDS by adopting an ensemble learning approach that combined AdaBoost with Boruta feature selection. They

implemented dimensionality reduction techniques such as PCA to address the challenges of IoT-based smart environments. This approach aimed to improve detection performance through optimized feature selection and reduced data redundancy. The model achieved high accuracy and low false alarm rates on benchmark datasets like NSL-KDD and BoT-IoT. By removing outliers and identifying the most relevant features, the model enhanced computational efficiency and reduced both training and prediction durations. The integration of CatBoost improved the model's ability to detect anomalies in highly imbalanced datasets which made it suitable for real-time applications in IoT networks. Despite its strengths, the research faced challenges in generalizing results across diverse IoT environments due to the limited representation of threats in the datasets used.

The research by *Ahmed, Beyioku & Yousefi (2024)* has examined the integration of honeypot data with machine learning to enhance cyber-attack detection in smart city IoT environments. Using high-interaction honeypots deployed over extended periods, the study captured authentic real-world attack data with an ambition to improve relevance and reliability. Several algorithms—naive Bayes, decision tree, K-nearest neighbor (KNN), sequential neural network (SNN), and LSTM—were evaluated against IoT-specific attacks. The key contributions of this framework included a comprehensive assessment of feature selection techniques, particularly Information Gain and One Rule (OneR), which improved model efficiency. Decision tree and LSTM models achieved high accuracy, with LSTM excelling in identifying temporal patterns critical for intrusion detection. In context of limitations of this research, the challenges included inconsistent performance from naive Bayes and missing data requiring preprocessing. Although feature selection enhanced performance, our assessment revealed that maintaining efficiency with larger datasets remained difficult. Moreover, the study emphasized thoroughly the limited availability of diverse public datasets and the complexity of scaling detection systems across heterogeneous IoT architectures. This phenomenon has highlighted the intense need for future work in honeypot-driven machine learning approaches for IoT security.

*Shareef et al. (2024)* introduced an IDS that combined the Zebra Optimization algorithm (ZOA) for feature selection with a dual-channel graph attention network (DGAN) for classification. This system addressed the structural and semantic challenges of IoT communications by incorporating node and semantic attention networks to identify intricate patterns in device interactions. Hyperparameter optimization using the Sooty Tern Optimization algorithm (STOA) further enhanced detection accuracy, achieving 99.87 percent and surpassing traditional models. The research demonstrated the ability to process large and noisy datasets while improving feature extraction efficiency. Temporal and relational patterns in real-time traffic were also effectively captured by this system. The author emphasized high classification precision and adaptability to evolving botnet behaviors that was achieved through the tailored use of deep learning architectures for IoT traffic. Nonetheless, the reliance on computationally intensive architecture posed scalability issues in resource-constrained IoT environments. Therefore, this study stressed the need for future research to optimize the balance between detection accuracy and system efficiency. Hereby, the 'Table 1' presents a comparative analysis of recent intrusion detection models designed for smart city infrastructure, focusing on scalability, real-time

**Table 1 Comparison of recent smart city IDS models (*Hazman et al., 2024*; *Almasri & Alajlan, 2023*; *Taher et al., 2023*; *Manickam et al., 2023*; *Hazman et al., 2023*; *Ahmed, Beyioku & Yousefi, 2024*; *Shareef et al., 2024*).**

| Model (Ref) | Properties (Scalability, Real-time, efficiency, Adaptability) | Techniques used (Algorithms, Methods) | Basic primitives (Datasets, classification approach, Evaluation metrics) | Scalability (Large-scale readiness) | Real-time processing | Optimization methods (Pruning, FS, etc.) | Dataset source | Performance metrics (Accuracy, precision, recall, F1, AUC) |
|---|---|---|---|---|---|---|---|---|
| *Hazman et al. (2024)* | High accuracy and low false alarms; handles high-dimensional, imbalanced IoT data; optimized for efficiency (TPU acceleration)–suitable for large-scale, real-time deployment | Deep LSTM network combined with Autoencoder-based feature extraction and Genetic Algorithm + Information Gain for feature selection. | Evaluated on multiple IoT intrusion datasets (BoT-IoT, Edge-IIoT, NSL-KDD) in both binary and multi-class modes; used accuracy, precision, recall as metrics (focused on minimizing FPR/FNR). | Yes–Designed for large-scale IoT networks; uses TPU hardware to handle high data volumes (though model generalization to all IoT environments is a noted challenge). | Yes–Capable of real-time detection due to greatly reduced training/inference time (TPU acceleration). | Dimensionality reduction *via* Autoencoders; feature selection with GA and IG to optimize input features; TPU-based acceleration for faster processing (no explicit pruning/quantization reported). | BoT-IoT, Edge-IIoT, NSL-KDD (public IoT/ICS intrusion datasets) | ~99% accuracy (high precision & recall) with negligible FPR/FNR on all tested datasets. |
| *Almasri & Alajlan (2023)* | Two-phase detection & mitigation system; offers real-time threat response with high accuracy and low false positives; adaptable (fuzzy logic adjusts to traffic changes); some computational overhead (heavy model) noted for resource-limited devices | Cascaded Adaptive Neuro-Fuzzy Inference System (CANFIS) for attack detection, followed by modified Deep Reinforcement Learning (MDRL) to isolate/block compromised devices. Also employs sequential feature selection during CANFIS phase to refine detection rules | Evaluated on two datasets: an IoT network intrusion dataset and the ISCX-2012 dataset; supervised classification of normal *vs.* attacks (comparisons made to LSTM, SVM, standard DRL); metrics include accuracy, precision, recall, F1-score | Moderate–Demonstrated generalizability by testing on multiple datasets, but the model's high memory/CPU requirements could hinder deployment at IoT edge scale (cloud resources likely needed). | Yes–Achieves significantly reduced detection time for immediate threat response (meets real-time detection needs in testing). | Feature selection (sequential) integrated to optimize CANFIS detection accuracy; combination of neuro-fuzzy and deep RL provides adaptive learning (no special pruning/quantization noted beyond architectural optimization). | Custom IoT intrusion traffic dataset + ISCX 2012 (benchmark IDS dataset) | Accuracy = 98.7%, outperforming baseline LSTM, SVM, and single-phase DRL models; high precision, recall, F1 (>0.95 reported) achieving low false-positive rate. |

(Continued)

| Model (Ref) | Properties (Scalability, efficiency, Real-time, Adaptability) | Techniques used (Algorithms, Methods) | Basic primitives (Datasets, classification approach, Evaluation metrics) | Scalability (Large-scale readiness) | Real-time processing | Optimization methods (Pruning, FS, etc.) | Dataset source | Performance metrics (Accuracy, precision, recall, F1, AUC) |
|---|---|---|---|---|---|---|---|---|
| *Taher et al. (2023)* | Focused on IIoT botnet detection; robust to high-dimensional data and diverse attacks (detects known and unknown botnet activities); improves efficiency by removing redundant features; suitable for real-time use in resource-constrained environments (fast convergence). | Hybrid feature selection FGOA-kNN (Fisher Score + Grasshopper Optimization + k-NN) to identify relevant features; optimized neural network (IHHO-NN) fine-tuned *via* improved Harris Hawks Optimization; uses chaotic maps and opposition-based learning to avoid local minima and speed up training. | Evaluated on N-BaIoT dataset (IoT device botnet traffic); multiclass classification of various botnet attack types *vs.* normal behavior; evaluated by accuracy, precision, recall–significantly higher than conventional classifiers (SVM, *etc.*) on this dataset. | Good–Effective in complex, high-volume IIoT data (demonstrated on large feature space). Nevertheless, reliance on a single dataset means broader scalability/ generalization to all IoT scenarios is unproven; the optimization algorithms add overhead that could tax extremely constrained devices. | Yes–Emphasizes real-time applicability; fast convergence and reduced feature set make it viable for timely detection in IIoT networks. | Bio-inspired optimization throughout: Grasshopper Optimization for feature selection, improved HHO for network parameter tuning; chaotic initialization and Random Opposition-Based Learning enhance search efficiency. (No specific model pruning mentioned, but feature elimination serves to streamline computation.) | N-BaIoT (Network-Based IoT) botnet dataset. | Outperformed baseline methods on N-BaIoT– higher accuracy, precision, recall (exact values not given, but substantially improved detection rates); low false alarm rate due to optimized feature set and tuning. |
| *Manickam et al. (2023)* | Anomaly detection model for IoT smart cities; achieves very high accuracy (~99%); captures temporal patterns *via* RNN; reduces false positives. Yet, it relies on centralized data processing, which may introduce latency and limit efficiency at scale; adaptability across highly diverse IoT setups not fully verified. | Binary Pigeon Evolution Optimization (BPEO) for feature selection; Elman Recurrent Neural Network (ERNN) for anomaly detection (sequence learning); Billiard Based Optimization (BBO) for hyperparameter tuning of the deep model. | Evaluated on benchmark IoT intrusion dataset (s) (not explicitly named in summary); supervised classification distinguishing normal *vs* malicious events; main evaluation metric is accuracy (along with false positive rate reduction). | Limited–While the approach is effective on the test dataset, the need to funnel data to a central node can become a bottleneck in large-scale deployments (network congestion, latency). Model generalization to other IoT datasets/ environments remains a concern due to possible overfitting to specific data. | Not explicitly–The centralized architecture could impede real-time performance in expansive networks (due to processing delays). In smaller-scale scenarios or with sufficient infrastructure, it can operate with low latency, but real-time capability isn't a primary proven feature. | BBO meta-heuristic optimizes the RNN's hyperparameters; BPEO selects a minimal relevant feature subset to reduce complexity. These optimizations jointly maximize detection accuracy and efficiency. (No mention of model pruning or quantization.) | Not specified (likely standard IDS datasets such as NSL-KDD, KDD99 or IoT-23.) | Accuracy >99% on test data; low false-positive rate achieved (high precision) due to effective pattern learning and feature selection |
| *Hazman et al. (2023)* | Ensemble-based IDS for smart environments; high accuracy and low false alarm rates on tests; improved computational efficiency *via* aggressive feature reduction and outlier removal (faster training/inference); suitable for real-time IoT deployment. Generalization may be limited by the narrow range of attacks in training data. | Ensemble learning with AdaBoost (and CatBoost) classifiers, combined with Boruta feature selection and PCA for dimensionality reduction. Data pre-processing included removing outliers to improve model focus. This pipeline optimizes the feature set and leverages boosting to handle class imbalance. | Evaluated on NSL-KDD and BoT-IoT datasets; performed binary/ multi-class classification of normal *vs* various attacks; measured by accuracy and false alarm rate (and likely precision/ recall). The inclusion of CatBoost aimed at maintaining performance on imbalanced data. | Moderate–By reducing data dimensionality and focusing on key features, the model is more scalable to larger datasets than unoptimized approaches. Nonetheless, its true scalability across different smart city setups is uncertain, as the training data covers limited threat types (potentially requiring retraining for new attack patterns). | Yes–The optimizations (feature reduction, faster ensemble learning) make it feasible for real-time intrusion detection in IoT networks, with quick model responses and manageable computational load. | Boruta algorithm and PCA prune irrelevant and redundant features, reducing complexity; outlier removal streamlines the training data. Uses ensemble boosting (AdaBoost + CatBoost) to optimize learning especially on imbalanced classes. No specific mention of network pruning/ quantization, focusing instead on input feature optimization. | NSL-KDD (classic IDS dataset) and BoT-IoT (IoT botnet traffic). | High accuracy (near 99%) on both datasets with very low false positive rates; improved precision/ recall in detecting attacks due to feature optimization and ensemble methods (exact metrics not given, but false alarms notably reduced) |

| Model (Ref) | Properties (Scalability, efficiency, Real-time, Adaptability) | Techniques used (Algorithms, Methods) | Basic primitives (Datasets, classification approach, Evaluation metrics) | Scalability (Large-scale readiness) | Real-time processing | Optimization methods (Pruning, FS, etc.) | Dataset source | Performance metrics (Accuracy, precision, recall, F1, AUC) |
|---|---|---|---|---|---|---|---|---|
| Ahmed, Beyioku & Yousefi (2024) | Leverages *real-world* IoT attack data *via* honeypots, improving authenticity of detection; evaluated multiple ML algorithms for intrusion detection; feature selection enhanced efficiency. Highlights challenges in consistency (some classifiers like NB underperformed) and data quality (missing values) for IoT security. Primarily an offline analysis (not yet optimized for live deployment). | Deploys high-interaction honeypots over an extended period to gather genuine IoT attack traffic. Applies various machine learning classifiers – Naive Bayes, Decision Tree, k-NN, Sequential NN, LSTM – to detect attacks. Utilizes Information Gain and OneR feature selection to identify important features and improve model accuracy/efficiency. | Dataset is a custom IoT attack dataset derived from the honeypot logs (captures real attack attempts on IoT devices). Framed as supervised classification of malicious *vs* normal events; evaluated using accuracy (and implicitly precision/recall *via* analysis of false positives/negatives). Decision Tree and LSTM showed consistently high accuracy in identifying IoT-specific attacks. | Limited–While using real attack data increases relevance, the approach is tested on a specific collected dataset. Scaling this method to a city-wide IoT deployment (with continuous data streams from many honeypots or live traffic) is non-trivial, as noted by the need for more diverse attack data and improved handling of large data volumes. | No (offline)–The study analyzes stored attack data rather than performing live monitoring. Real-time detection in practice would require addressing the processing of streaming data and model deployment on IoT infrastructure, which the authors note as a challenge (due to data volume and varying performance of algorithms). | Feature selection (IG, OneR) is used to reduce feature space and improve classification speed/accuracy. Otherwise, standard ML optimizations per algorithm were applied; no custom pruning or model compression beyond data preprocessing. | IoT honeypot-derived dataset (authentic attack traffic collected from deployed traps). | High accuracy achieved by Decision Tree and LSTM models (best performers) in detecting captured attacks. NB was less effective. Precise metrics not given in summary, but results indicate strong detection capability for DT/LSTM (likely >90% accuracy) and improved efficiency after feature selection. |
| Shareef et al. (2024) | Graph neural network-based botnet detector for IoT; extremely high accuracy (99.87%) and precision; captures complex structural and temporal patterns in network traffic *via* dual attention mechanisms; adaptable to evolving botnet behaviors. Computationally intensive architecture (Graph Attention Network + optimization algorithms) may impede deployment on constrained IoT devices | Zebra Optimization Algorithm (ZOA) for feature selection (identifies salient network traffic features); Dual-channel Graph Attention Network (DGAN) for classification, with separate node-level and semantic attention sub-networks to learn intricate communication patterns; Sooty Tern Optimization (STOA) for hyperparameter tuning of the DGAN. | Evaluated on a large-scale IoT botnet traffic dataset with substantial noise and diversity (dataset name not given, likely a recent IoT/CPS dataset). Performs binary classification (botnet *vs* normal traffic) with graph-based approach. Key metrics: accuracy = 99.87%, with very high precision and strong recall (surpassing traditional IDS models). | Mixed–Capable of handling large, noisy data inputs (demonstrated on big dataset), indicating good data scalability. Though, the method's heavy computational demands raise scalability issues for real-world deployment: it may not scale down well to many low-power devices without further optimization. Likely suited for centralized analysis or powerful edge servers in a smart city. | Partial–The system is designed to analyze streaming IoT traffic and did capture real-time patterns effectively in testing. Yet, due to its complexity, achieving low-latency inference on typical IoT hardware is challenging; real-time performance is attainable only if sufficient computational resources are available. | Feature selection *via* ZOA reduces input complexity; STOA tunes hyperparameters for optimal model performance. The dual-attention GNN architecture itself is an optimization for capturing both structural and temporal features of traffic. No mention of model pruning/quantization–the focus is on algorithmic optimization (bio-inspired search for features and params). | Large IoT network traffic dataset with botnet attacks (name not specified). | Accuracy = 99.87%; very high precision (near-perfect classification performance) reported, with strong recall and AUC presumably close to 1.0 given the accuracy. Significantly outperforms conventional IDS benchmarks. |

**Table 2 Feature comparison of IoT datasets.**

| Dataset | Year | Attack types | Features | Devices | Real/ Sim | Labelled | Related works reference |
|---|---|---|---|---|---|---|---|
| MedBIoT (*Hao et al., 2024*) | 2020 | IoT botnets (Mirai, Bashlite, Torii–causing DDoS, C&C traffic) | 115 (Network flow stats) | Mixed IoT (83 devices) | Real | Yes | *Ahmed, Beyioku & Yousefi (2024)* |
| IoT-23 (*Sharma & Babbar, 2024*) | 2020 | Various IoT malware (20 captures incl. Mirai variants, C&C traffic, DDoS) | 20+ (Zeek flow fields) | Various IoT (Raspberry Pi + real IoT devices) | Real | Yes | *Ahmed, Beyioku & Yousefi (2024)* |
| BoT-IoT (*Alosaimi & Almutairi, 2023*) | 2018 | DDoS, DoS, scanning (Reconnaissance), keylogging & data exfiltration (Theft) | 47 (Extracted flow features) | Smart home | Sim | Yes | *Hazman et al. (2024, 2023)* |
| Edge-IIoT (*Nuaimi et al., 2023*) | 2022 | 14 IoT/IIoT attacks in five categories (DoS/DDoS, information gathering, injection, MITM, malware) | 84 (Selected features from 1,176) | Industrial | Sim | Yes | *Hazman et al. (2024)* |
| NSL-KDD (*Zakariah et al., 2023*) | 2009 | Classic attacks (DoS, R2L, U2R, Probe) | 41 (Connection features) | General network | Sim | Yes | *Hazman et al. (2024, 2023)* |
| IoT network intrusion (Smart Home) (*Kaur et al., 2023*) | 2019 | Various (*e.g.*, host scan, botnet malware, MITM, DDoS) | 115 (46 (Traffic features)) | Smart home | Real | Yes | *Almasri & Alajlan (2023)* |
| ISCX 2012 (*Shiravi et al., 2012*) | 2012 | Multi-stage attacks (SSH brute force, HTTP DoS/DDoS, infiltration) | 20 (flow metrics) | General network | Real | Yes | *Almasri & Alajlan (2023)* |
| N-BaIoT (*Naveed, 2020*) | 2018 | IoT botnet malware (Mirai, BASHLITE– multiple attack vectors) | 115 | Smart home | Real | Yes | *Taher et al. (2023)* |
| CIC-IoT 2023 (*Canadian Institute for Cybersecurity, 2023*) | 2023 | 33 large-scale IoT attacks in seven classes (DDoS, DoS, Recon, Web, Brute Force, Spoofing, Mirai) | ~80 (Network-flow features, CSV) | Diverse IoT (105 devices) | Real | Yes | – |
| ACI-IoT 2023 (*Army Cyber Institute, 2023*) | 2023 | Scanning (Reconnaissance), flooding (DoS), password cracking (Brute Force), ARP spoofing | NetFlow records (*e.g.*, 15 fields) | Smart home IoT (lab setup) | Real | Yes | – |

processing, and efficiency in handling IoT threats. Each model is evaluated based on its core techniques, optimization strategies, dataset sources, and performance metrics to highlight advancements and practical applicability in securing IoT-based environments.

To further strength the argument, we have also compiled Table 2, which provides a detailed comparison of key IoT datasets, outlining their characteristics and differences to aid in identifying features suited to the proposed methodology.

# PRELIMINARIES

## Vulnerability context for smart city networks

Smart city infrastructures are attractive targets for cyberattacks due to the convergence of critical services and interconnected devices that make it crucial to understand the potential attack vectors, targets, and impacts. Prominent attack vectors include DDoS which

overwhelms systems with excessive traffic, Command and Control (C&C) exploits that manipulate devices for malicious activities, and Advanced Persistent Threats (APTs) which involve prolonged infiltration to compromise network security and disrupt essential services. These attacks often target critical infrastructure such as power grids and emergency services, aiming to cause significant disruptions and compromise public safety. Botnets, a key component in many attacks, are networks of compromised devices controlled by attackers. They can be structured in centralized, decentralized, or hybrid architectures, each influencing their resilience and control mechanisms. Malware employed in these attacks includes polymorphic malware that changes its code to evade detection, and techniques like cross-layer exploits that target multiple layers of a system. Infection vectors, such as vulnerability exploitation, supply chain compromise, and social engineering, are used to infiltrate devices. Attackers also utilize defense evasion techniques to conceal their malicious activities.

## Challenges in IoT botnet detection

Our investigation revealed several limitations and challenges, one of the major challenges in IoT botnet detection is the extreme heterogeneity of IoT devices, which introduces a wide variety of communication protocols, processing capabilities, and security vulnerabilities across networks. Each IoT device can behave differently depending on its configuration which proves its' eligibility in creating a highly diverse attack surface that complicates the modeling of normal and malicious behavior. Another significant challenge arises from the resource constraints inherent in IoT devices, such as limited computational power, memory, and energy. These limitations obstruct the deployment of advanced security measures such as encryption and real-time anomaly detection. As a result, IDS must depend on lightweight solutions that maintain a balance between detection accuracy and computational efficiency. IoT devices generate a substantial volume of network traffic due to their continuous operation. This results in massive data streams that must be processed and analyzed in real time. It was evident that high data throughput demands real-time detection capabilities, which places a significant computational load on detection systems. These systems must operate efficiently to handle the traffic without causing latency. A further challenge lies in obtaining labeled datasets necessary for training machine learning models. IoT botnets often exhibit subtle behavioral changes that are difficult to distinguish from legitimate traffic. Herewith, the absence of comprehensive labeled datasets limits the effectiveness of supervised learning methods in detecting emerging threats. Likewise, modern IoT botnets add to this complexity by frequently adapting their attack strategies to evade detection mechanisms. Techniques such as encryption, polymorphism, and traffic obfuscation make IDS ineffective when relying solely on signature-based or traditional anomaly detection methods.

## Traditional IDS limitations

Traditional IDS face several limitations (*Hajiheidari et al., 2019*; *Khraisat & Alazab, 2021*; *Najafli, Haghighat & Karasfi, 2024*) when applied to modern IoT environments. For instance:

a. It lacks the capability to manage the evolving and diverse nature of IoT devices and protocols. This results in significant gaps in detecting targeted and complex attacks.

b. High false positive rates are a common issue due to rigid rule-based detection methods that fail to adapt to varying traffic patterns.

c. These systems often exhibit poor scalability in large-scale IoT environments. This leads to performance degradation and delays in threat detection.

d. Limited real-time analysis capabilities arise because majority of traditional IDS rely on batch processing methods, which are unsuited for the continuous flow of IoT data.

e. Resource-constrained IoT devices are incompatible with traditional IDS due to the high computational demands of these systems.

f. Signature-based detection methods in traditional IDS cannot identify novel or previously unknown threats effectively.

g. Contextual awareness is insufficient in traditional IDS. They fail to recognize the holistic behavior of IoT devices within their operational settings, resulting in incomplete assessments of threats.

h. Integration challenges occur with IoT-specific communication protocols and encryption mechanisms, leading to undetected vulnerabilities in traffic analysis.

## Features of ideal DL architectures for intrusion detection

Our investigation revealed the following advantages of optimum DL for intrusion detection (*Najafli, Haghighat & Karasfi, 2024*; *Muneer et al., 2024*), which includes, but not limited to:

a. Ability to extract features from raw network traffic directly from data without requiring manual feature engineering. This ensures suitability for high-dimensional IoT datasets.

b. The capability to identify and interpret intricate patterns within data that enable detection of sophisticated and evolving attack strategies is often undetected by traditional methods.

c. High adaptability to diverse IoT devices and communication protocols. This ensures robustness across heterogeneous network environments.

d. Scalability that allows effective handling of large data volumes. This makes these methods ideal for real-time intrusion detection in smart city networks.

e. Potential to significantly reduce false positives by identifying subtle traffic deviations. This improves alerting precision and operational effectiveness.

f. Resilience against adversarial tactics through advanced modeling of non-linear relationships and the integration of diverse data sources.

g. Ability to perform well even with limited labeled data by employing semi-supervised or unsupervised approaches. This addresses challenges in scenarios where labeled datasets are scarce.

## Applied preprocessing and optimization techniques

To ensure the integrity, efficiency, and effectiveness of the proposed intrusion detection framework, a combination of data preprocessing and model optimization techniques was employed throughout the design and training phases. Each technique was selected to address specific challenges associated with high-dimensionality, data sparsity, computational efficiency, and model generalization in considered network environment. The following definitions provide concise explanations of the methods utilized within this study to support feature representation, model training, and real-time inference:

→ **Min-max normalization:** A scaling technique that transforms numerical features into a fixed range, typically [0, 1], to ensure uniform contribution to the model's learning process.

→ **Mutual information and variance thresholding techniques:** Feature selection methods used to retain informative attributes and eliminate low-variance or weakly relevant features which usually leads to reducing redundancy in high-dimensional IoT traffic data.

→ **Imputation techniques:** Methods applied to handle missing data by substituting null values with statistical estimates, such as mean for numerical and mode for categorical features, to maintain dataset consistency.

→ **Feature scale normalization:** A transformation approach that adjusts the magnitude of numerical features to a common scale, preventing dominant attributes from skewing model training.

→ **One-hot encoding:** A categorical encoding scheme that converts non-numeric variables into binary vectors which allow the models to process protocol types and device states without assuming ordinal relationships.

→ **Weight decay:** A regularization technique that penalizes large weight values during training to prevent overfitting and promote model generalization in noisy IoT environments.

→ **L2 regularization:** A specific form of weight decay that adds the squared magnitude of weights to the loss function to discourage complexity and improve robustness.

→ **Dropout and early stopping:** Regularization strategies where dropout randomly disables neurons during training and early stopping halts training once validation performance ceases to improve and reduce overfitting.

→ **Pruning optimization:** A technique that removes non-contributing or weakly active neural connections to reduce model size and improve inference speed without compromising detection accuracy.

→ **Weight quantization:** The process of converting model weights from high-precision to lower-precision formats (*e.g.*, 32-bit to 8-bit) to decrease memory footprint and accelerate computations.

→ **Sparse matrix multiplication:** An inference-time optimization that leverages zero-valued weight sparsity to skip unnecessary computations to enhance real-time performance on edge devices.

# PROPOSED METHODOLOGY

It is evident that the presence of botnets in IoT-equipped smart city networks can lead to disastrous consequences, undermining the functionality and security of critical infrastructure systems. These networks, designed to manage everything from traffic lights and public transportation to utilities and public safety systems, are particularly vulnerable to botnet attacks, which can harness compromised IoT devices to launch coordinated disruptions. Such attacks can cripple urban operations, cause widespread service outages, and expose sensitive public and personal data.

## Data collection and preprocessing

Raw network traffic data were collected from a diverse set of IoT devices operating within the smart city infrastructure, including but not limited to smart meters, environmental sensors, traffic monitoring systems, and surveillance cameras. These devices were selected due to their heterogeneous communication protocols (*i.e.*, includes but not limited to: Zigbee, Wi-Fi, LoRaWAN, RTSP (Real-Time Streaming Protocol), *etc.*,) and varying data generation rates, representing the complexity of an interconnected smart city ecosystem. The collection process spanned over a period of 4 weeks to ensure temporal diversity and capture data variations in network behavior. Data capture was conducted using network monitoring tools (*i.e.*, Wireshark & TCPDUMP) capable of passive traffic analysis, which allowed for continuous recording without interfering with device operations. Packet sniffers (*i.e.*, as illustrated in Fig. 1) were deployed at multiple network access points across the emulated smart city setup to capture all inbound and outbound traffic to ensure that the dataset reflected both normal and anomalous behavior from different sectors of the emulated city. This approach allowed for the monitoring of device-level communications, network congestion patterns, and potential cyber threats such as DDoS attempts, while maintaining data integrity. To ensure comprehensive representation, network traffic from high-traffic nodes such as public Wi-Fi hotspots was included alongside traffic from low-traffic IoT devices such as smart parking meters. This approach captured a broad spectrum of traffic patterns and device interactions. The collection framework incorporated various IoT protocols including MQTT, CoAP, and HTTP, which are integral to smart city infrastructure. Random sampling was applied across different time slots and network segments to avoid overrepresentation of any specific device type or network traffic pattern. In this context, we observed that data bias may arise from unequal representation of certain patterns or features within the dataset that can potentially lead to skewed or inaccurate model outcomes. Also, overrepresentation of traffic from high-traffic IoT devices like public Wi-Fi hotspots could hinder the detection of attacks targeting fewer common devices. Similarly, an imbalance in protocol representation, such as focusing on MQTT over HTTP was observed with the ability to reduce the model's capacity to detect threats in underrepresented communication types.

In our investigation, data preprocessing was essential for maintaining the quality and usability of collected IoT network traffic data for intrusion detection. We realized the importance of preserving the integrity and statistical reliability of the dataset, and for this reason, the mean imputation was used for numerical features by replacing null entries with

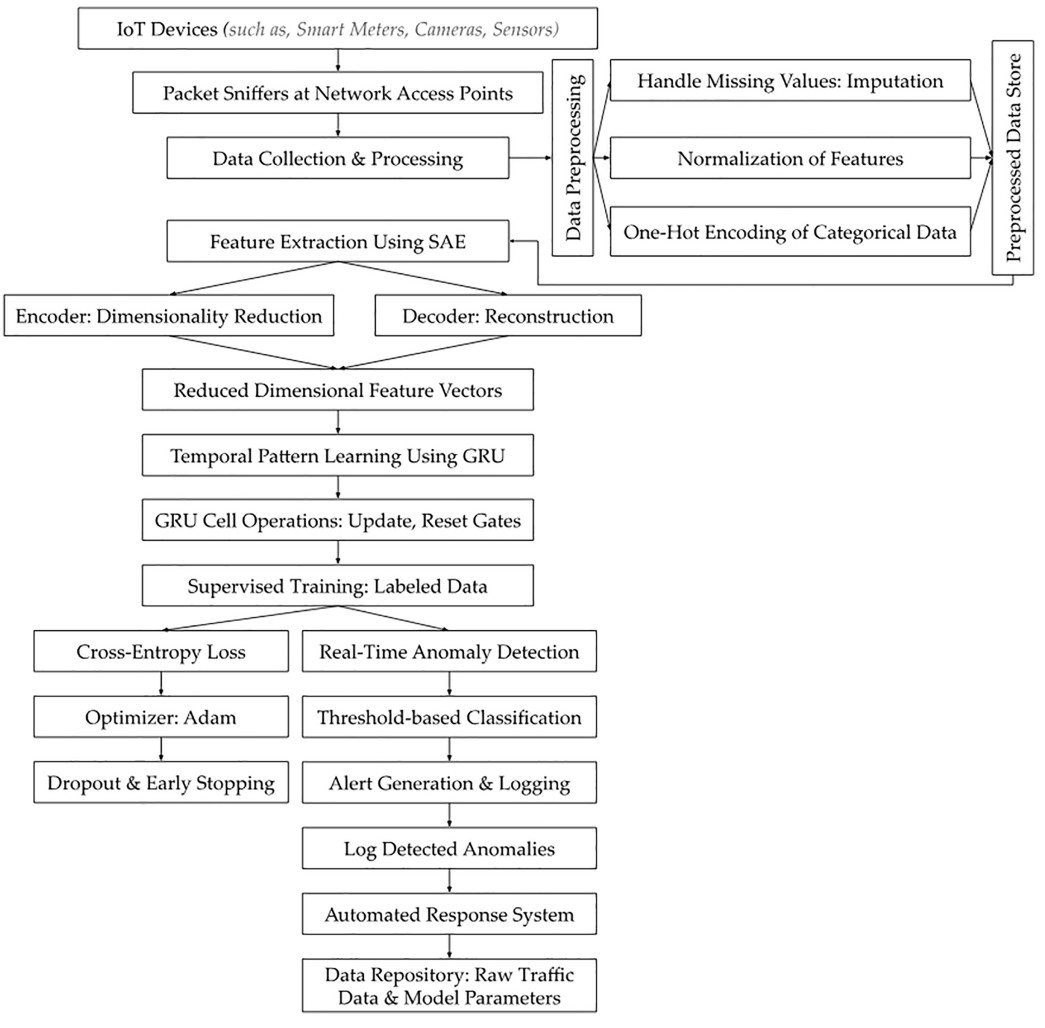

**Figure 1 SAE-GRU based intrusion detection system for IoT botnets.**

the arithmetic average of observed values, which preserved the statistical distribution and prevented distortion in features such as packet size, byte counts, and flow duration—attributes critical for distinguishing volumetric and behavioral anomalies in botnet detection.

To further strengthen the data preprocessing, we also applied Mode imputation to categorical features such as protocol type or device state, where the most frequent class was substituted to retain consistency in discrete attribute distribution. These strategies prevented information loss while maintaining model input uniformity. It is worth highlighting that for in botnet detection, numerical values are especially important as they capture subtle deviations in traffic volume, session timing, and data transfer behavior that often precede or accompany coordinated botnet activity, thus allowing the model to learn and identify patterns that are otherwise undetectable through categorical inspection alone.

**Table 3 List of smart city equipped IoT devices employed to generate the dataset.**

| IoT device name | Vendor | Purpose | Specifications | Communication protocols | Vulnerabilities |
|---|---|---|---|---|---|
| Smart meter | Siemens (INHEM1216) | Monitor and report energy usage | *Communication:* ZigBee, *Protocol:* IEC 62056, *Power:* Low Power (Battery) | ZigBee, LoRaWAN | Data tampering, Unauthorized access |
| Traffic monitoring camera | Bosch (MIC inteox 7100i) | Track traffic flow and violations | *Resolution:* 1080p, *Connectivity:* 4G/5G, *Power:* Wired | 4G/5G, Wi-Fi | Eavesdropping, DDoS |
| Environmental sensor | Honeywell (C7355A1050) | Monitor air quality and environmental conditions | *Sensor Type:* Temp./Hum. /$CO_2$/TVOC/ PM, *Connectivity:* LPWAN, *Power:* Solar/Battery | LPWAN, MQTT | Spoofing, Data tampering |
| Smart streetlight | GE Lighting (ERL1-ERLH-ERL2) | Control and manage street lighting | *Connectivity:* LoRaWAN, *Power:* Solar, *Control:* Remote Dimming | LoRaWAN | Unauthorized access, Remote exploitation |
| Public Wi-Fi hotspot | Cisco (240AC, Catalyst IW9165E) | Provide public wireless internet access | *Standard:* IEEE 802.11ac, *Range:* 90 m, *Power:* Power over Ethernet (PoE) | Wi-Fi, IEEE 802.11ac | Man-in-the-Middle (MITM), Data Interception |
| Smart parking meter | Presto (Presto 1000), ParkingBOXX | Monitor parking spaces and payment systems | *Communication:* NB-IoT, *Payment:* Contactless, *Power:* Battery | NB-IoT, LTE | Spoofing, Data exfiltration |
| Surveillance drone | DJI (Mavic Pro Platinum) | Provide aerial surveillance | *Range:* 10 km, *Camera:* 4K, *Power:* Rechargeable Battery | 4G/5G, Wi-Fi | GPS Spoofing, Signal jamming |
| Smart waste management Bin | Bigbelly (Smart) | Manage waste levels in urban areas | *Sensor:* Ultrasonic, *Communication:* LoRa, *Power:* Solar | LoRaWAN | Denial of service, Sensor spoofing |
| Smart water monitoring system | Xylem (DIQ/S 281–WTW, pH 298-WTW) | Monitor water levels and quality | *Sensor Type:* pH, Turbidity, *Connectivity:* LTE-M, *Power:* Battery | LTE-M | Data exfiltration, Device hijacking |

In accordance, the feature selection improved model efficiency by eliminating irrelevant attributes while preserving essential patterns in network behavior.

We also applied mutual information along with a fixed variance thresholding technique set at 0.01 to eliminate features exhibiting low variability across observations, as such features contribute minimal discriminative power and can introduce noise into the learning process. This dual-step approach enabled the selection of highly informative attributes that not only improve classification accuracy by focusing the model on relevant patterns but also reduce computational complexity by removing redundant or static input dimensions. The applied dataset covered various intrusion types, including DDoS attacks, Command and Control activities, traffic injection attempts, data exfiltration, spoofing events, malware propagation, credential harvesting, brute force attacks, firmware exploitation, and eavesdropping on communication channels.

The heterogeneous nature of IoT devices, as presented in Table 3, introduced variations in data formats, requiring specific methods to handle missing values. In this context, imputation techniques were applied to maintain dataset integrity, where mean imputation

was used for numerical features and mode imputation for categorical attributes to address data sparsity. Whereas the feature scale normalization was applied to account for differences in magnitude, including packet sizes and traffic volumes. Min-max normalization adjusted feature values within a unified scale, typically between 0 and 1, which guaranteed consistency across the dataset and improving the effectiveness of model training.

$$x_{norm} = \frac{x - x_{min}}{x_{max} - x_{min}}. \tag{1}$$

As per Eq. (1), the $x_{norm}$ is normalized value, $x$ is the original feature value, $x_{min}$ & $x_{max}$ are the minimum and maximum values of the feature, respectively. This normalization is crucial in preventing features with larger ranges from dominating the training process.

To convert categorical variables, such as protocol types, into numerical formats, one-hot encoding (*Rezvan et al., 2024*) was employed. This method transformed each categorical feature into a binary vector, where each category was represented as a distinct dimension that has allowed the model to interpret the categorical information without imposing artificial ordinality. The choice of one-hot encoding was particularly relevant given the non-hierarchical nature of many categorical features in IoT traffic, where no intrinsic ranking exists between different protocols or device types.

We selected preprocessing algorithms such as the Pandas and Scikit-learn libraries due to their proven efficiency and scalability. These qualities were critical for handling the extensive volume and high dimensionality of IoT data. The preprocessing steps (*such as:* Missing value imputation; Normalization of feature scales; One-hot encoding for categorical variables; Data cleaning; Standardization; Feature extraction; and Feature dimensionality reduction) standardized the dataset while eliminating inconsistencies. This process ensured a robust foundation for feature extraction and effective model training.

## Rationale for employing stacked autoencoder

The adoption of SAEs in this research stems from their capacity to process high-dimensional data effectively while maintaining essential features. The emulated IoT environment generated extensive network traffic with a mix of significant and redundant information. Hereby, the SAE proved effective in this context due to its ability to uncover latent patterns such as recurring traffic bursts, synchronized connection attempts, and consistent session durations by encoding these into compact, information-rich representations (*e.g.*, protocol activity footprint; connection uniformity pattern, *etc.*). Through dimensionality reduction, the autoencoder transformed correlated features: like 'packet size, flow duration, and byte count' into a unified latent variable capturing their combined variance. This abstraction enabled the model to detect subtle yet coordinated anomalies in communication behavior that are often obscured in the original high-dimensional feature space especially for those which are indicative of botnet activity. This compression process minimized noise and preserved critical characteristics without compromising analytical depth (*i.e.*, feature importance analysis; performance metrics evaluation; temporal pattern recognition; model generalization assessment; real-time

detection capability; false positive and false negative analysis; cross-validation; and dataset diversity evaluation). In our resource-constrained IoT system, this dimensionality reduction demonstrated its value by enabling computational efficiency and facilitating real-time analysis. The compressed data produced by SAEs reduced the computational demands of subsequent tasks, thereby it improved the overall performance of implemented intrusion detection model.

In the proposed emulated settings, SAE, the encoder function transforms the high-dimensional input data $x \in R^n$ into a lower-dimensional hidden representation $h \in R^m$. This procedure (*i.e.*, as exhibited in Eq. (2)) was important to reduce redundancy and emphasize essential feature interactions which enabled the model to focus on patterns that are most relevant for distinguishing between normal and malicious traffic behavior.

$$h = f(W_e x + b_e) \tag{2}$$

where $W_e \in R^{m \times n}$ is the weight matrix, $b_e \in R^m$ is the bias term, and $f(\cdot)$ is the activation function, typically ReLU (Rectified Linear Unit). The hidden representation $h$ encapsulates the essential features in a compressed form. The decoder then attempts to reconstruct the input $x$ from $h$ using the Eq. (3):

$$x' = f(W_d h + b_d) \tag{3}$$

where $W_d \in R^{n \times m}$ and $b_d \in R^n$ are the decoder's weight matrix and bias term, respectively. The objective of training the SAE is to minimize the reconstruction error, which measures the difference between the original input $x$ and its reconstruction $x'$. The reconstruction error is often expressed as the mean squared error (MSE) and is minimized as follows (*i.e.*, expressed in Eq. (4)):

$$L(x, x') = \frac{1}{n} \sum_{i=1}^{n} (x_i - x_i')^2 \tag{4}$$

where $L(x, x')$ is the loss function, and $x$ and $x'$ represent the original and reconstructed values of the input data, respectively. It is worth noting that minimizing the MSE during the training of the SAE ensured that the reconstructed output closely matches the original input for preserving critical traffic patterns. Lower MSE indicates that the encoder has captured the most informative features needed to differentiate normal from botnet-induced anomalies. Thus, an encoder compresses input data into a lower-dimensional space and a decoder reconstructs the original input. The objective is to retain the most relevant features while discarding noise which facilitates efficient pattern recognition in high-dimensional datasets.

The SAE was trained in an unsupervised manner by optimizing the loss function using the gradient-based optimization algorithm 'Adam' (*Reyad, Sarhan & Arafa, 2023*). This algorithm was selected for training the SAE because it combined the benefits of both 'AdaGrad and RMSProp (*i.e.*, are the optimization algorithms used to update model parameters during training)' by adaptively adjusting learning rates for each parameter using first and second moment estimates of gradients. This made it well-suited for handling sparse features and non-stationary objectives common in high-dimensional IoT

traffic data. Regularization technique, weight decay or *L2* regularization, was applied to the weight matrices during training to mitigate overfitting. Our investigation revealed that this approach is particularly critical when handling large and noisy datasets that are common in IoT networks. Weight decay introduced a penalty to the loss function based on the magnitude of the weights. This penalty helped prevent the model from becoming overly complex and improved its ability to generalize to unseen data. The applied loss function with *L2* regularization is expressed in Eq. (5) as:

$$L_{reg}(x, x') = L(x, x') + \lambda \, \|W\|_2^2 \tag{5}$$

where $\lambda$ is the regularization parameter and $\|W\|_2^2$ is the squared *L2* norm of the weight matrices.

After training the SAE, the encoder component processed high-dimensional input data to generate lower-dimensional feature representations. These compact feature vectors retained the most critical information for subsequent analysis while minimizing computational complexity in later stages of intrusion detection. This reduction in dimensionality enhanced computational efficiency and concentrated on the most crucial patterns which eventually led to improved accuracy in identifying malicious activities.

## Underlying principle for exercising gated recurrent unit network

The GRU network was implemented to detect *temporal patterns* (*e.g.*, repeated connection intervals, synchronized packet bursts, periodic data uploads, uniform session durations, cyclic protocol switching, timed beacon signals, consistent idle-to-active transitions, recurring port access sequences, repetitive handshake patterns, scheduled command transmissions) in IoT network traffic data. Herewith, the structured gated recurrent units handled *sequential data* (*i.e.*, time-stamped network flows, session-wise traffic logs, packet arrival sequences, protocol exchange orders, connection attempt histories, authentication sequences, port scanning timelines, device communication intervals, traffic burst timings, malware propagation traces) efficiently by maintaining long-term dependencies while reducing computational demands. Temporal patterns and sequential data were important because they captured the timing and order of network activities, which are critical for identifying behaviors characteristic of botnet operations. Such patterns revealed coordinated actions, delayed triggers, and repetitive access sequences that static feature analysis could not detect.

Our comprehensive evaluation showed that GRUs address vanishing gradient challenges that frequently affect traditional recurrent neural networks. This property strengthens their effectiveness in real-time intrusion detection for IoT environments. Cross-validation confirmed the capability of GRUs to capture sequential dependencies with greater stability, whereas conventional recurrent networks struggled to retain long-range information. Thus, this implementation enabled the emulated system to learn patterns efficiently and identify both normal and malicious activities within the traffic. Normal activities include regular data transmission intervals, periodic updates, scheduled maintenance tasks, consistent communication behaviors, synchronized sensor readings, uniform packet sizes, and steady bandwidth utilization. Whereas, the malicious activities

encompassed sudden surges in traffic volume, irregular packet timings, anomalous data flows, unscheduled device interactions, inconsistent protocol behaviors, rapid data bursts, significant packet size deviations, coordinated surges across multiple devices, repeated connection attempts to external servers, frequent contact with command-and-control servers, brute force login attempts, high-frequency port scans, periodic data exfiltration bursts, prolonged DDoS attacks, unexpected increases in device-to-device communication, synchronized anomalies from geographically dispersed devices, abnormal encryption patterns, frequent traffic destination changes, and suspicious outbound connections to unverified domains. We have also observed that the GRUs require fewer parameters than LSTMs, demonstrated reduced computational overhead, which was vital in environments with limited resources. This efficiency and robust ability to process sequential dependencies established GRUs as an ideal solution for intrusion detection within the proposed architecture.

In our emulated settings, the functionality of a GRU (*i.e.*, as evident from Fig. 1) is controlled by two primary gates: the update gate and the reset gate, which govern how much past information is passed to the future. Herewith, the 'update gate' determines how much of the previously observed network behavior (*e.g.*, normal or suspicious traffic patterns) is carried forward for further analysis in the detection process. For botnet detection, this helped track long-term anomalies, such as coordinated attacks that evolve over time. Whereas the 'reset gate' decides how much of the past information should be ignored, enabling the model to focus on immediate and relevant traffic patterns. In our applied detection, this allowed the GRU to disregard irrelevant or benign variations, such as normal fluctuations in traffic, while concentrating on identifying emerging botnet behavior.

Accordingly, the update gate, denoted as $z_t$, determines the extent to which the hidden state from the previous time step $h_{t-1}$ should be carried forward. This phenomenon is represented in Eq. (6):

$$z_t = \sigma(W_z x_t + U_z h_{t-1} + b_z) \tag{6}$$

where $W_z$ and $U_z$ are the weight matrices for the input $x_t$ and the previous hidden state $h_{t-1}$, respectively, and $b_z$ is the bias term. The sigmoid function $\sigma$ ensures that the values of $z_t$ remain between 0 and 1, effectively controlling the weight of the previous hidden state in the current computation.

The reset gate, denoted as $r_t$, is responsible for deciding how much of the previous hidden state should be forgotten. As stated earlier, this gate plays a critical role in allowing the GRU to selectively reset portions of its memory when modeling sequential dependencies, as exhibited in Eq. (7):

$$r_t = \sigma(W_r x_t + U_r h_{t-1} + b_r). \tag{7}$$

Here, $W_r$ and $U_r$ are the weight matrices, and $b_r$ is the bias associated with the reset gate. By modulating the reset gate, the GRU can ignore irrelevant parts of the past sequence when they are not useful for future predictions. With this, as exhibited in Eq. (8), the candidate activation, denoted as $\widetilde{h}_t$, represents the new memory content to be added to the
hidden state. This candidate is computed based on both the current input and the reset hidden state, which allows the GRU to conditionally forget or remember parts of the sequence:

$$\widetilde{h}_t = \tanh(W_h x_t + U_h(r_t \odot h_{t-1}) + b_h. \tag{8}$$

Herewith, the $W_h$, $U_h$, and $b_h$ are the corresponding weights and biases for the candidate activation, and $\odot$ represents element-wise multiplication. The reset gate $r_t$ ensures that only relevant historical information influences the new candidate activation. Ultimately, the hidden state at time step $t$ is updated as a linear interpolation between the previous hidden state and the candidate activation, controlled by the update gate:

$$h_t = (1 - z_t) \odot h_{t-1} + z_t \odot \widetilde{h}_t. \tag{9}$$

Equation (9) determines the final output of the GRU unit at each time step by balancing the influence of the new candidate activation $\widetilde{h}_t$ with the previous hidden state $h_{t-1}$, depending on the value of the update gate $z_t$. This interpretation allowed GRUs to adaptively retain relevant information over long time sequences which no doubt proved to be a critical feature for identifying patterns of malicious activity within sequential IoT network traffic.

Herewith, the Table 4 provides an algorithmic and structured breakdown of the SAE-GRU-based intrusion detection workflow by detailing each step from data preprocessing to real-time threat classification. This formal representation enhances the clarity of the proposed methodology.

## Temporal pattern recognition in IoT network traffic

The compressed feature vectors generated by the SAE were passed as input into the GRU network to model temporal dependencies[3]. These lower-dimensional feature vectors, which represented condensed and denoised versions of the original high-dimensional network traffic, encapsulated critical information relevant to identifying both normal and malicious activities. The GRU network processed these sequential inputs by maintaining a hidden state that captured information from previous time steps. This mechanism allowed it to learn long-term dependencies and temporal correlations within the data. Each feature vector in the input sequence passed through the update and reset gates to enable the GRU to determine how much past information should be retained or discarded at each time step. Through this modulation, the GRU dynamically adjusted its memory of past events, which was essential for distinguishing between normal traffic patterns and anomalies indicative of botnet activity. As the GRU iterated through the sequence of feature vectors, it identified patterns that signaled malicious behaviors such as coordinated spikes in traffic or abnormal data flows while filtering out routine variations in the network. The ability of the GRU to maintain context over extended periods enabled it to detect botnet activities that unfold over long time frames. This temporal modeling was crucial for differentiating between benign and malicious behaviors, as it accounted for both short-term fluctuations and long-term trends in order to ensure high accuracy in threat identification.

[3] Temporal dependencies refer to the relationship between events occurring over time, where the current state of a system is influenced by past observations. In projected context of intrusion detection, temporal dependencies were essential for identifying patterns that evolve over multiple time steps to allow the model to distinguish between normal network behavior and malicious activity based on historical trends.

**Table 4 Algorithmic representation of the SAE-GRU based intrusion detection model.**

| Step | Algorithmic pseudocode |
|---|---|
| 1. Data collection and preprocessing | **Input:** Raw network traffic data $D$<br><br>**For** each network traffic record $X$ in $D$:<br><br>$\rightarrow$ Handle missing values using mean/mode imputation<br>$\rightarrow$ Normalize numerical attributes using Min-Max scaling<br>$\rightarrow$ Convert categorical variables using One-Hot Encoding<br>$\rightarrow$ Apply feature selection using Mutual Information & Variance Thresholding<br><br>**End For** |
| 2. Feature extraction using stacked autoencoder (SAE) | **Input:** Preprocessed network traffic features $X$<br>**Encode:** $Z = fW_eX + b_e$<br>**Decode:** $\hat{X} = f(W_dZ + b_d)$<br>**Compute Loss:** $(L)$ |
| 3. Temporal pattern recognition using GRU | **Input:** $Z$ from SAE<br>**Initialize:** GRU hidden state $h_0$<br>**For** each time step $t$<br><br>$\rightarrow$ Compute update gate: $z_t = \sigma(W_zX_t + U_zh_{t-1} + b_z)$<br>$\rightarrow$ Compute reset gate: $r_t = \sigma\left(W_rX_t + U_rh_{\{t-1\}} + b_r\right)$<br>$\rightarrow$ Compute candidate activation: $\widetilde{h_t} = tanh\left(W_hX_t + U_h\left(r_t.\,h_{\{t-1\}}\right) + b_h\right)$<br>$\rightarrow$ Update hidden state: $h_t = (1 - z_t).h_{\{t-1\}} + z_t.\tilde{h}_t$<br><br>**End For**<br>**Output:** Final hidden state $h_t$ for classification |
| 4. Intrusion classification using thresholding | **Input:** GRU output $h_t$ Compute classification score $S$<br>**If** $S > \tau$ then:<br><br>$\rightarrow$ **Classify as Malicious Else**<br>$\rightarrow$ **Classify as Normal End If** |
| 5. Optimization for real-time processing | **Apply:**<br><br>$\rightarrow$ Model pruning to remove redundant parameters<br>$\rightarrow$ Weight quantization to reduce precision for faster inference<br>$\rightarrow$ Sparse matrix multiplication for efficient computation<br>$\rightarrow$ Deploy model on edge nodes for low-latency detection |
| 6. Alert generation and response | **If** attack is detected:<br><br>$\rightarrow$ Generate alert with timestamp, device ID, attack type<br>$\rightarrow$ Log alert in central database<br>$\rightarrow$ Trigger security response: block IP, isolate device, deploy monitoring<br><br>**End If** |
| 7. Model training and performance evaluation | **Train model:**<br><br>$\rightarrow$ Minimize loss function: $L = -\sum ylog(\hat{y})$<br>$\rightarrow$ Use Adam optimizer for learning rate adjustment<br>$\rightarrow$ Apply dropout regularization & early stopping<br>$\rightarrow$ **Evaluate model:** Compute **Accuracy, Precision, Recall, F1-score, AUC** |

## Supervised training process of the GRU network

The supervised training process of the GRU network was conducted using labeled data, where each sequence of input feature vectors derived from the SAE was associated with a corresponding class label indicating whether the traffic pattern represented normal behavior or malicious activity. The use of labeled data allowed the GRU to learn from ground-truth examples and distinguish between benign and anomalous patterns in network traffic. The cross-entropy loss function was employed during training to quantify the difference between the predicted class probabilities and the actual class labels. This loss function was suitable for binary classification tasks, such as detecting botnet attacks, and is represented in Eq. (10):

$$L(y, \hat{y}) = -\frac{1}{N} \sum_{i-1}^{N} [y_i \log(\hat{y}_i) + (1 - y_i)\log(1 - \hat{y}_i)] \tag{10}$$

where $y_i$ is the true label, $\hat{y}_i$ is the predicted probability, and $N$ represents the number of samples. Herewith, we applied the Adam optimizer to compute individual adaptive learning rates for each parameter and combines the benefits of both the AdaGrad and RMSProp algorithms (as exhibited in Table 5), ensuring stable convergence even in complex, high-dimensional spaces.

To enhance the generalization capability of the GRU network and prevent overfitting, two techniques were applied: Dropout and Early Stopping. Dropout was introduced during training by randomly setting a fraction of the units in the hidden layers to zero at each iteration. This forced the network to learn redundant representations and reduced its dependency on specific neurons. The dropout rate was carefully selected to achieve a balance between underfitting and overfitting, typically ranging from 0.3 to 0.5 for optimal performance. Early Stopping was employed as a regularization method that monitored the model's performance on a validation set and halted training when the validation loss stopped improving. This ensured that the GRU network stopped training at its optimal state and avoided overfitting the training data. Our observation revealed that by applying these strategies, the GRU network achieved high accuracy and robustness in detecting anomalies across diverse and complex IoT traffic patterns and demonstrated strong generalization to unseen data during deployment.

## Architecture of the combined SAE-GRU model

From the preceding discussion, it is evident that we have effectively conceptualized & favorably designed optimal SAE-GRU model to efficiently process high-dimensional IoT network traffic data and extracted temporal dependencies for effective intrusion detection. Herewith, at this stage, the SAE acted as a dimensionality reduction layer to reduce noise and compressed the raw high-dimensional input data into lower-dimensional feature vectors. These feature vectors are then passed into the GRU network, which models the temporal relationships between the sequences of traffic data. We designed the SAE as a three-layer architecture: an input layer, a hidden layer, and an output (reconstruction) layer. The number of hidden units in each layer was empirically determined, with the

**Table 5 Execution of AdaGrad and RMSProp in IoT botnet detection.**

| | |
|---|---|
| **AdaGrad** | **1. Initialize:**<br><br>+ Set initial learning rate $\eta$.<br>+ Initialize gradient accumulator $\boldsymbol{g} = \boldsymbol{0}$.<br><br>**2. For each parameter $\theta_t$ at time $\boldsymbol{t}$:**<br><br>+ Compute gradient $\nabla\theta t$ based on current traffic data.<br>+ Accumulate squared gradients: $\boldsymbol{g} - \boldsymbol{g} + (\nabla\theta_t)^2$.<br>+ Adjust learning rate: $\eta_t = \frac{\eta}{\sqrt{g}+\varepsilon}$ (where $\epsilon$ is a small constant to prevent division by zero).<br>+ Update parameter: $\theta_{t+1} = \theta_t - \eta_t\nabla\theta_t$.<br><br>**3. End For.** |
| **RMSProp** | **1. Initialize:**<br><br>+ Set initial learning rate $\eta$.<br>+ Initialize moving average $g = 0$, set decay rate $\rho$.<br><br>**2. For each parameter $\theta_t$ at time $\boldsymbol{t}$:**<br><br>+ Compute gradient $\nabla\theta_t$ based on current traffic data.<br>+ Update moving average of squared gradients: $g = \rho g + (1 - \rho)(\nabla\theta_t)^2$.<br>+ Adjust learning rate: $\eta_t = \frac{\eta}{\sqrt{g}+\epsilon}$ (where $\epsilon$ prevents division by zero).<br>+ Update parameter: $\theta_{t+1} = \theta_t - \eta_t\nabla\theta_t$.<br><br>**3. End For.** |

Herewith,

➢ η: The *initial learning rate*, which controls how much to adjust the model's weights in response to the estimated error during training.

➢ $\Theta_t$: The *model parameter* at time step $\boldsymbol{t}$, representing the weights of the network that are being updated.

➢ $\nabla\theta_t$: The *gradient* of the loss function with respect to the parameter $\theta_t$, representing the direction and rate of change to reduce the error.

➢ $\boldsymbol{g}$: The *gradient accumulator* in AdaGrad or the *moving average* of squared gradients in RMSProp. It stores the sum or exponential decay of squared gradients to adapt the learning rate over time.

➢ $\epsilon$: A *small constant* added to avoid division by zero during the learning rate adjustment. This ensures numerical stability when gradients are very small. Numerical stability is crucial in emulated botnet detection because the learning process for detecting botnet patterns involved processing large amounts of real-time, high-dimensional network traffic data. In this contexts, small gradients caused learning rates to become unstable, potentially leading to either overly large updates (which would cause the model to diverge) or excessively small updates (which might slow down or halt learning). Thus during our assessment, it was evident that if numerical stability is not maintained, the model exhibited: (a) *Fail to converge* (*i.e.*, without stability, updates to the model parameters became erratic, making it difficult for the model to reliably detect botnet behavior), (b) *Misclassify traffic patterns* (*i.e.*, unstable learning resulted in poor detection of subtle or evolving botnet activities, leading to false negatives (missing attacks) or false positives (flagging benign traffic as malicious), and (c) *Cause computational inefficiencies* (*i.e.*, unstable calculations did lead to inefficiencies or even system crashes, delaying the real-time detection needed to respond to botnet threats).

➢ $\eta_t$: The *adjusted learning rate* at time step $\boldsymbol{t}$, modified by the accumulated or moving average of the squared gradients.

➢ $\rho$: The *decay rate* in RMSProp, controlling the moving average calculation by determining how much of the previous squared gradient information is retained.

➢ $\theta_{t+1}$: The *updated parameter* at time step $\boldsymbol{t + 1}$, after applying the learning rate adjustment and gradient-based correction.

hidden layer of the encoder having 128 units to balance expressiveness and computational efficiency. The rectified linear unit (ReLU) was used to introduce non-linearity and enhance the model's ability to learn complex patterns. The GRU component comprised

two layers of recurrent units, each with 64 hidden units, and uses the tanh activation function[4], which provided stability in gradient propagation, reduced the risk of exploding or vanishing gradients, especially when processing long sequences. The choice of this architecture ensured that the model could capture both spatial and temporal features in network traffic data while remaining computationally feasible for real-time deployment.

The deployment of the trained SAE-GRU model for real-time intrusion detection in the emulated network was achieved using Python-based machine learning libraries (*i.e.*, TensorFlow), which supported efficient implementations of deep learning architectures. The model was integrated into the existing network monitoring system through RESTful APIs to enable real-time data stream input and analysis. Real-time inference was optimized by batching incoming network packets and processed them through the SAE for compression followed by GRU-based temporal analysis. The system operated on a cluster of edge computing nodes equipped with moderate GPU support, which distributed the computational workload of IoT traffic processing across multiple nodes. This hardware configuration ensured that the model fulfilled the strict low-latency requirements of real-time anomaly detection.

The mechanism for anomaly detection relied on analyzing incoming data streams and comparing them to normal behavior patterns learned during training. Once a sequence of compressed feature vectors passes through the GRU, the model outputs a score representing the likelihood of the sequence being normal or malicious. Herewith, the Thresholding method was applied to classify network behavior. A decision threshold $T$ is empirically set by analyzing the Receiver Operating Characteristic (ROC) curve during validation. The value of $T$ is chosen to balance the trade-off between false positives and false negatives, minimizing the total classification error. If the output score $s(x)$ exceeds $T$, the behavior is classified as malicious. This is represented in Eq. (11):

$$Class = \begin{cases} Malicious, & if \quad s(x) > T \\ Normal, & if \quad s(x) \leq T. \end{cases} \tag{11}$$

The model also incorporated a mechanism to handle false positives and false negatives by maintaining an alert threshold buffer, which reduced the likelihood of triggering false alarms due to benign fluctuations in network traffic. Upon detection of suspicious patterns indicative of botnet activity, the SAE-GRU model automatically triggered a sophisticated alert mechanism that was designed to efficiently inform network administrators and initiate response protocols. The alert mechanism was integrated directly into the IDS in order to ensure swift and effective responses.

Herewith, when the SAE-GRU model identifies an anomaly that exceeds the predefined decision threshold $T$, it triggers an alert generation process. This process involves the formation of an alert packet that includes detailed information about the detected anomaly, such as the time of detection, affected network segments, and a risk assessment score. This packet is then communicated to the IDS through a secure communication channel, ensuring that the information is relayed promptly and securely to the network administrators.

[4] The *tanh activation function* was employed in the GRU network to maintain stable gradient propagation during the processing of sequential data and to address the vanishing gradient problem. Its capacity to map inputs within the range of −1 to 1 allowed the GRU model to capture both positive and negative temporal dependencies, which made it effective in detecting intricate patterns in network traffic.

**Table 6 Automated response decision.**

Input: Anomaly type, Risk score

Output: Mitigation action

begin

 if Risk score > High risk threshold then

   Execute high-priority response protocol

    *Actions:* Immediate isolation of affected network nodes, automated blocking of suspected malicious IP addresses, rapid deployment of autonomic security patches to vulnerable systems, forceful termination of unauthorized connections, real-time alerts to security operation centers.

 else if Risk score > Medium risk threshold then

   Execute Moderate-priority response protocol

    *Actions:* Enhanced monitoring of suspected traffic, temporary restriction of network access privileges for suspicious accounts, updating autonomically the firewall rules to restrict unusual traffic patterns, automatically conducting vulnerability scans on potentially affected segments, initiating detailed forensic analysis for gathered intelligence.

 else

   Execute low-priority monitoring protocol

    *Actions:* Logging detailed event information for future analysis, regular updates of anomaly signature definitions, execution of performance and security audits on potentially impacted systems, passive monitoring of network traffic for emerging patterns.

 end if

end

The implemented alert mechanism included an automated logging system that records every detected event into a centralized log database. This database is structured to store comprehensive details of all alerts, facilitating subsequent analysis and forensic investigations. The stored data includes timestamps, sensor IDs, the type of detected anomalies, severity levels, and the actions taken in response to the alerts. This logging is crucial for tracking the effectiveness of the detection system, auditing system responses, and refining detection strategies over time.

The applied IDS was programmed to parse incoming alerts and categorize them based on severity. Depending on the severity and the specific characteristics of the detected anomaly, predefined mitigation strategies are automatically initiated. These strategies include, but are not limited to, autonomic reconfiguring firewalls to block malicious traffic, segmenting parts of the network to isolate compromised devices, and deploying additional monitoring to the affected areas/zones. The system employs a rule-based decision algorithm, as outlined in Table 6, to determine the appropriate response based on the risk assessment score and the type of anomaly detected.

With this, each action was logged with a corresponding entry in the incident management system, which included timestamps, the nature of the response, and status updates on the resolution of the issue. This system not only ensured immediate attention to potential threats but also allowed the designated network administrators to review and adjust the automated responses based on effectiveness and evolving network security requirements.

## Model optimization

In our emulated assessment, optimizing the SAE-GRU model was crucial to ensure efficient performance while maintaining detection accuracy. One of the primary optimization techniques applied was model pruning. Pruning involves systematically removing less important neurons or connections from the model, which reduced the overall complexity without significantly compromising its ability to detect anomalies. By identifying and eliminating parameters that contributed minimally to the output during the training phase, we reduced both the storage requirements and inference time. This approach was particularly effective for the GRU component, where certain recurrent connections were pruned based on their contribution to the loss function.

Weight quantization was another technique applied to reduce the computational load. This method involves converting high-precision floating-point weights into lower-precision formats, such as 8-bit integers. While this results in a minimal loss of precision, it significantly decreases the memory footprint and increases the speed of inference on edge devices that often lack powerful processing units. In this consideration, the 'Quantization-aware training[5]' was used to account for the reduced precision during the training phase which guaranteed that the model remained robust despite the compressed representation of weights.

To further enhance efficiency, we implemented the Sparse matrix multiplication algorithm (*i.e.*, as exhibited in Fig. 2) designed to minimize computational overhead. Specifically, sparse matrix operations were employed during inference, allowing the model to bypass unnecessary computations associated with zero-valued parameters after pruning. This was complemented by batch processing, where multiple incoming data streams were processed simultaneously to utilize the parallelism to reduce latency. For temporal data processing in the GRU, we employed truncated backpropagation through time (TBPTT) to limit the number of sequential time steps during training. This reduced the computational demand by restricting the depth of the network's temporal memory to make it suitable for real-time applications where long sequences would otherwise introduce delays.

To further elaborate on the applied optimization techniques, pruning in the proposed model was not executed as a one-time static reduction but dynamically adjusted during training using a gradient sensitivity score. Parameters with persistently low gradient contributions across epochs were identified and removed, especially within GRU gates where inactive pathways were filtered out to enhance information flow. For weight quantization, the approach involved a mixed-precision strategy where core arithmetic layers operated on 8-bit integers while retaining 16-bit accumulators in sensitive layers to preserve gradient fidelity. This reduced computational overhead without undermining convergence behavior. Herewith, the Sparse matrix multiplication was applied with structure-aware indexing where zero-valued weights were skipped using pre-computed mask arrays to permit optimized memory access patterns and cache-friendly execution. Applied technique exploited sparsity resulting from pruning to accelerate matrix-vector operations in the recurrent layers. These optimizations were tailored for execution on edge

[5] *Quantization-aware training* was utilized to account for reduced precision during the training phase, ensuring that the model remained robust when deploying weights in lower-precision. This approach minimized the loss of accuracy while reducing the computational load, making the model suitable for real-time intrusion detection in resource-constrained emulated IoT environment.

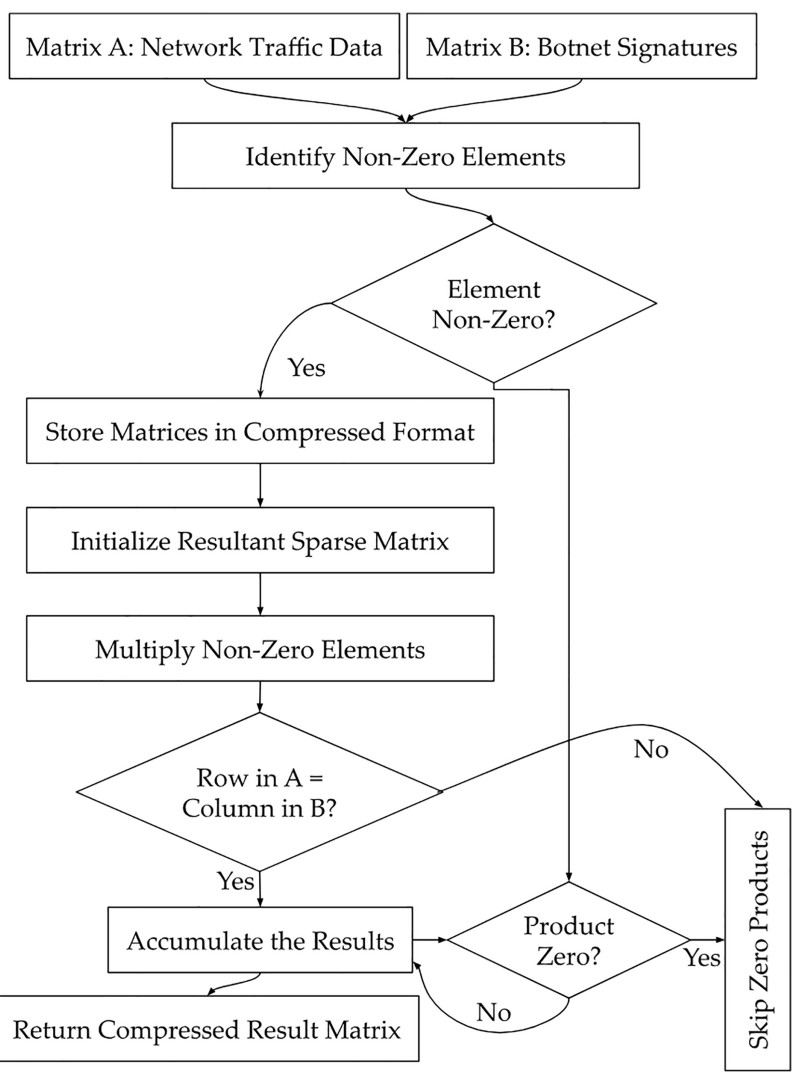

**Figure 2 Sparse matrix multiplication to minimize computational overhead.**

devices with constrained compute profiles so that the model would become eligible to meet latency constraints under real-world traffic load.

For accurate representation, it is essential to convey that the Sparse matrix multiplication was not merely used as a computational convenience but was structurally embedded into inference routines to minimize multiplications involving zero-valued weights post-pruning. This choice directly reduced processing latency, particularly during peak network loads, where real-time response is critical. Correspondingly, quantization-aware training allowed floating-point weights to be compressed into lower-bit representations while retaining learning capacity through adjusted gradient computations. By simulating inference-time quantization during training, the model achieved reliability, consistency, and robustness in edge deployments. We also tailored the

model pruning by using a relevance-based thresholding mechanism, which systematically removed parameters with minimal impact on classification loss, especially within the GRU gates where redundant recurrence was detected. The model employed 'truncated backpropagation through time' to handle long sequences efficiently to enable temporal learning with reduced memory consumption. We also applied the batch processing across parallel IoT streams which further exploited hardware concurrency that allowed the model to meet sub-second decision latency.

The cumulative effect of optimization technique ensured that the SAE-GRU model could be deployed in metropolitan IoT environments with limited processing and memory resources, while it is still capable of providing real-time, high-accuracy anomaly/malware/intrusion detection. By balancing computational efficiency and detection performance, the model effectively addressed the unique challenges posed by IoT-based smart city networks, where rapid and lightweight processing is paramount for maintaining network security.

## EMULATION RESULTS AND DISCUSSION

The emulation setup for the projected research had been designed to replicate a comprehensive and realistic smart city network environment with diverse IoT devices that has ensured a robust framework for testing the proposed intrusion detection system. The experimental hardware included high-performance servers powered by Intel Xeon processors with 32 cores and 64 threads on each server. These specifications supported parallel processing required for handling large volumes of network traffic and running machine learning models. Each server was equipped with 128 GB of DDR4 RAM to handle memory-intensive tasks such as feature extraction and deep learning training efficiently. The storage infrastructure consisted of a combination of 1 TB NVMe SSDs for fast data access and 10 TB HDDs for long-term data retention. To enable high-bandwidth communication, each server had been fitted with 10 Gbps Network Interface Cards to ensure minimal latency during traffic processing and real-time anomaly detection. The networking infrastructure included enterprise-grade switches and routers, which had been configured to simulate the interconnectivity typical of a smart city. Wireless access points supporting 802.11ac and 5G had been deployed to mimic public and private IoT communication layers. The IoT devices, as detailed in Table 3, included a range of environmental sensors, smart meters, and surveillance cameras.

The software environment had also been carefully tailored with the servers running a combination of Ubuntu Linux for backend operations and Windows Server for data management tasks. IoT devices were configured to operate on lightweight systems (*i.e.*, FreeRTOS) for low-power sensors and customized Android OS for more advanced devices like smart meters. Network emulation was conducted using Mininet, which enabled the creation of a virtual network topology comprising hundreds of simulated IoT devices. Mininet had been configured to mimic various smart city infrastructure components including public Wi-Fi hotspots and smart utility management system. The emulated cloud environment was hosted on Amazon EC2 (*AWS, 2024*) to allow scalability for testing cloud-based services and data management (*Liu, Wang & Liu, 2023*). Traffic from IoT

devices was programmed using Python and C++ (pseudocode and partial dataset (*Tariq & Ahanger, 2024*)) to maintain real-time control over device behavior and communication patterns.

Furthermore, the emulation configuration was designed with a multi-layered network topology, comprising more than 200 nodes to represent different types of IoT devices and network nodes. The link characteristics were configured to simulate real-world conditions, with bandwidths ranging from 10 Mbps for low-power IoT devices to 1 Gbps for critical infrastructure nodes like public Wi-Fi hotspots. Latency varied between 5 and 200 ms depending on the distance between nodes and the type of communication being emulated. Packet loss rates were intentionally varied between 0% and 5% to study the system's resilience under different network conditions. Traffic was generated using 'iperf and TC (traffic control)' utility to emulate different traffic patterns, such as constant bit rate for routine operations and bursty traffic during periods of high demand. Multiple attack scenarios, including DDoS, botnet propagation, and malware injection, were emulated to stress-test the system. Each attack was configured with varying intensities and durations to assess how the model adapts to different attack vectors to ensure a thorough evaluation of the intrusion detection system's robustness.

Network analyzers such as Wireshark and tcpdump were used to monitor behavior and collect data from the emulated network. These tools captured real-time traffic from IoT devices and saved packet data for analysis. To track server performance, tools (*i.e.*, iftop and htop) monitored CPU load, memory usage, and network bandwidth during attack simulations. Custom Python scripts automated the collection and organization of network metrics, providing a clear view of both normal and malicious traffic patterns.

For data visualization and analysis, PowerBI was employed to generate time-series graphs and heatmaps that allowed for a clear understanding of traffic patterns and anomaly detection outcomes. This tool was used to provide real-time dashboards that displayed key performance metrics, including attack detection rates, false positive rates, and network throughput during the experiments. The collected data was analyzed using a combination of Python's statistical libraries (NumPy and SciPy) and machine learning framework (*i.e.*, TensorFlow and PyTorch) to ensure that the detection model was evaluated with high precision. This combination of tools provided comprehensive insights into the system's performance which enabled us to effectively evaluatively the SAE-GRU's ability to detect and respond to various forms of IoT botnet attacks.

## Performance evaluation

As it is evident from the preceding section's discussion, the evaluation methodology used to assess the performance of the SAE-GRU model was rigorous and multifaceted. It aimed to provide a thorough understanding of its effectiveness in detecting IoT botnet activities. For evaluation, we employed metrics such as accuracy, precision, recall, F1-score, and the area under the receiver operating characteristic curve (AUC). These metrics were selected

[6] False positive rate measures the proportion of benign instances incorrectly classified as threats, which is a critical factor in evaluating the reliability of applied IDS. A low false positive rate reduces unnecessary alerts, ensuring that security mechanisms focus on actual threats rather than misclassifications.

for their ability to measure the correctness of the model's predictions while balancing the detection of true positives against the minimization of false positives[6]. Thus:

a. Precision measured the proportion of true positive detections among all positive predictions, ensuring that false positives (benign traffic misclassified as malicious) are minimized.

b. Recall, on the other hand, measured the proportion of actual attacks that were correctly detected, which is critical for ensuring that no malicious behavior goes undetected.

c. The F1-score provided a harmonic mean between precision and recall which offered a balanced metric in scenarios where both false positives and false negatives are of concern.

d. The area under the ROC curve (AUC) was used to evaluate the model's ability to distinguish between normal and malicious traffic over a range of thresholds that provided a more nuanced view of the classifier's performance.

Hence, the accuracy of the model is defined as Eq. (12):

$$Accuracy = \frac{TP + TN}{TP + TN + FP + FN} \tag{12}$$

where $TP$ represents true positives $TN$ true negatives, $FP$ false positives, and $FN$ false negatives. Accuracy is a simple metric to assess overall performance but can be misleading in imbalanced datasets. To address this, precision is defined as Eq. (13):

$$Precision = \frac{TP}{TP + FP}. \tag{13}$$

Recall, which reflects the proportion of true positives among all actual positives, is expressed as: Eq. (14):

$$Recall = \frac{TP}{TP + FN}. \tag{14}$$

The F1-score combines precision and recall into a single metric, especially useful in scenarios where both false positives and false negatives carry significant costs. It is defined as:

$$F1 - score = 2 \times \frac{Precision \times Recall}{Precision + Recall}. \tag{15}$$

Eventually, the AUC measures the model's ability to distinguish between classes (malicious *vs.* normal traffic) across different thresholds, with values closer to '1' indicating better performance.

Hereby, Table 7 presents the performance evaluation of the proposed model using ten-fold cross-validation to demonstrate its effectiveness in detecting IoT botnet activities. The results indicate that the model achieves consistently high accuracy, precision, recall, and F1-score across multiple iterations that is indicative of its robustness in differentiating between normal and malicious network traffic. The low false positive rate confirms the

**Table 7 Performance metrics of SAE-GRU intrusion detection model in IoT botnet detection across ten-fold cross-validation.**

| Iteration# | Accuracy (%) | Precision (%) | Recall (%) | F1-Score (%) | AUC |
|---|---|---|---|---|---|
| 1 | 98.5 | 97.8 | 98.2 | 98.0 | 0.992 |
| 2 | 98.7 | 98.1 | 98.5 | 98.3 | 0.993 |
| 3 | 98.6 | 98.0 | 98.3 | 98.1 | 0.992 |
| 4 | 98.8 | 98.3 | 98.6 | 98.4 | 0.994 |
| 5 | 98.7 | 98.2 | 98.5 | 98.3 | 0.993 |
| 6 | 98.6 | 98.0 | 98.4 | 98.2 | 0.992 |
| 7 | 98.5 | 97.9 | 98.3 | 98.1 | 0.992 |
| 8 | 98.7 | 98.2 | 98.5 | 98.3 | 0.993 |
| 9 | 98.6 | 98.0 | 98.4 | 98.2 | 0.992 |
| 10 | 98.8 | 98.3 | 98.6 | 98.4 | 0.994 |
| Average | 98.65 | 98.11 | 98.43 | 98.27 | 0.993 |

model's ability to minimize erroneous classifications, which is essential for real-world deployment.

To establish a comprehensive and meaningful evaluation of the proposed SAE-GRU architecture, reference models *Hazman et al. (2024)* to *Shareef et al. (2024)* were carefully selected based on their methodological diversity, real-world applicability, and relevance to the evolving demands of intrusion detection in IoT-centric smart city environments. These approaches represent state-of-the-art strategies that utilize advanced deep learning, optimization algorithms, and hybrid techniques to address the complex landscape of IoT security. Models such as LSTM-AE with feature engineering (*Hazman et al., 2024*) and CANFIS-MDRL (*Almasri & Alajlan, 2023*) incorporate temporal modeling and adaptive inference, which are directly comparable to the GRU component of our architecture, making them suitable benchmarks for assessing temporal learning efficacy. Similarly, frameworks that use bio-inspired optimization—such as FGOA-kNN (*Taher et al., 2023*), BBO-ERNN (*Manickam et al., 2023*), and ZOA-DGAN (*Shareef et al., 2024*)—emphasize dimensionality reduction and adaptive feature selection, aligning closely with the SAE's role in our model. These models were evaluated using comparable datasets (*e.g.*, BoT-IoT, NSL-KDD, N-BaIoT, and IoT-23), which supports direct comparison across performance metrics like accuracy, recall, and AUC.

A 360-degree (*i.e.*, all inclusive, as exhibited in Table 7, and Figs. 3 to 8) performance evaluation was conducted using cross-validation technique to ensure the robustness and generalization of the SAE-GRU model. A k-fold cross-validation approach was applied, where the dataset was divided into $k$ subsets, and the model was trained on $k-1$ folds while the remaining fold was used for testing. This process was repeated k times to average the performance across all subsets to provide a comprehensively reliable estimate of the model's effectiveness. The model was tested on various datasets, including synthetic botnet traffic generated during the emulation and real-world IoT botnet datasets (*i.e.*, described in Table 2). The results were compared against baseline models, such as traditional machine

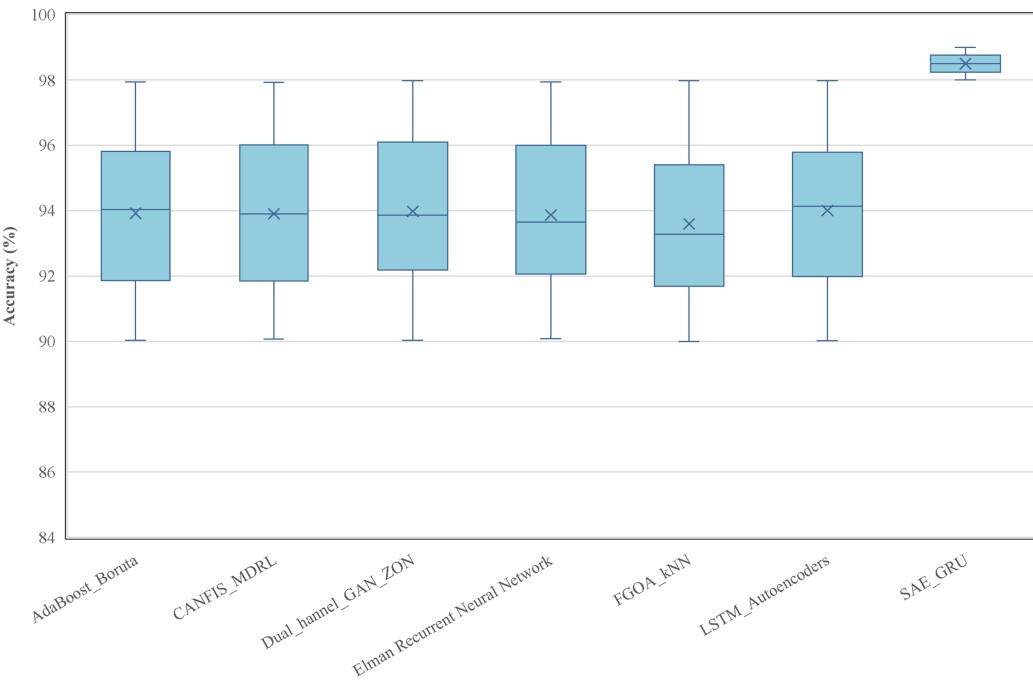

**Figure 3 Accuracy distribution across models.** This box plot illustrates the accuracy distribution across various machine learning models used for intrusion detection in IoT systems. Each box represents the interquartile range (IQR) with the horizontal line inside indicating the median accuracy. The whiskers extend to the minimum and maximum values within 1.5 times the IQR, while the cross symbols denote the mean accuracy. The SAE_GRU model shows the highest median and least variance, reflecting its consistent performance compared to other models such as AdaBoost_Boruta and CANFIS_MDRL, which display wider variability in accuracy.

learning-based IDS and deep learning models like LSTM networks, to validate the superior performance of the SAE-GRU model. For instance:

Figure 3 illustrates the accuracy distribution across various machine learning models for intrusion detection which highlights their performance variations. Among the models, SAE-GRU exhibits the highest median accuracy with minimal variance that indicates its robust and consistent performance. Other models demonstrate broader accuracy ranges, reflecting higher sensitivity to data variability or parameter configurations. The SAE-GRU model achieves better outcome due to its integration of Stacked Autoencoders for efficient feature extraction, gated recurrent units for capturing temporal patterns, and robust training strategies that enhance generalization while minimizing noise, overfitting, and computational complexity.

The evaluation of accuracy, precision, recall, and F1-score in Fig. 4 emphasized the robustness of SAE-GRU over alternative methodologies. By focusing on feature compression and temporal pattern recognition, SAE-GRU demonstrated reduced variance across performance metrics which indicates higher generalizability in real-world IoT systems.

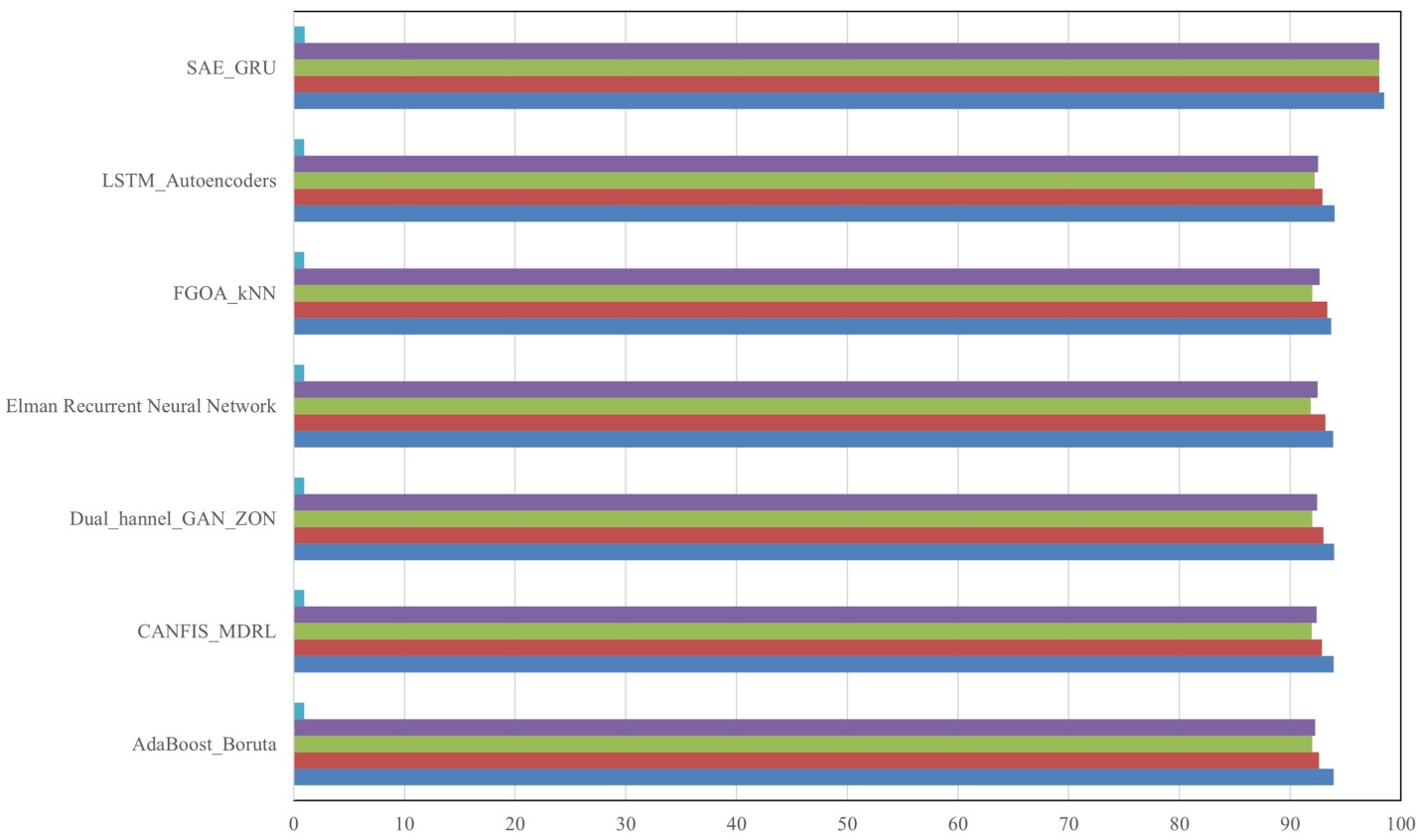

**Figure 4 Average of accuracy, precision, recall, and f1-score per model.** The average performance metrics—AUC, F1-score, recall, precision, and accuracy—across different machine learning models for intrusion detection. Each color represents a specific metric, with blue for AUC, purple for F1-score, green for recall, red for precision, and dark-blue for accuracy. SAE_GRU shows consistently high values across all metrics, indicating its superior performance compared to models like AdaBoost_Boruta and CANFIS_MDRL, which exhibit relatively lower and less consistent results.

Figure 5 highlighted the trade-offs between computational efficiency and scalability to showcase the ability of SAE-GRU to maintain optimal performance in resource-constrained environments through its dimensionality reduction and parallel processing capabilities. The scatter plot revealed that traditional models experienced significant scalability bottlenecks, whereas SAE-GRU effectively balanced throughput and real-time processing demands. Here, it is worth noting that the threshold values for classification were determined through empirical tuning using the receiver operating characteristic curve analysis where the optimal threshold was selected based on the highest true positive rate with the lowest false positive rate. A paired t-test (*i.e.*, comparison of the means of two related datasets to determine if there is a statistically significant difference between them) was conducted to compare the SAE-GRU model against baseline models, which demonstrated a statistically significant improvement in detection accuracy with a *p*-value below 0.05.

Insights into the sensitivity and specificity of models were revealed by mapping (*i.e.*, as illustrated in Fig. 6) false positive and false negative rates. The SAE-GRU model achieved

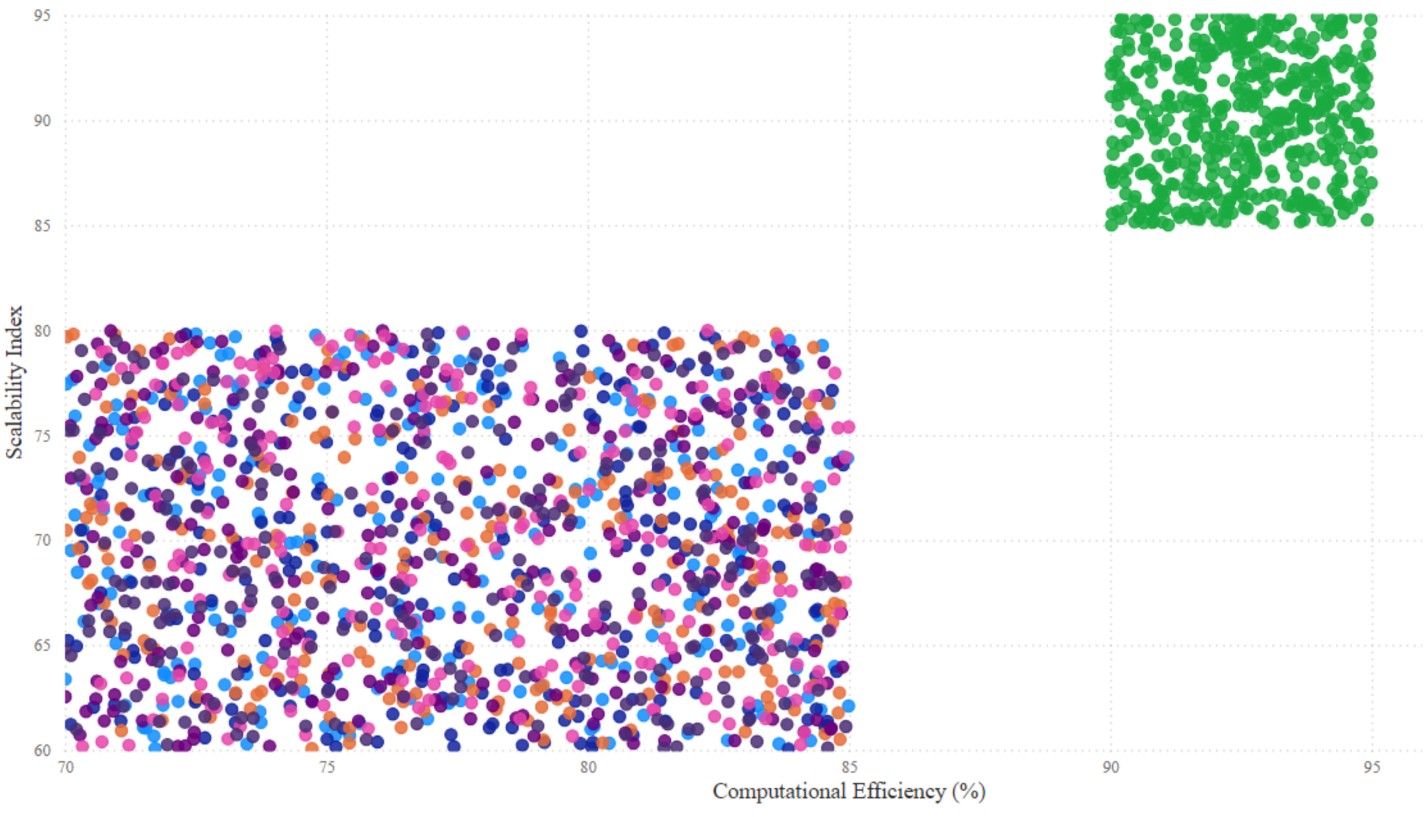

**Figure 5 Computational efficiency *vs*. scalability index.** The relationship between computational efficiency and scalability index for different machine learning models, with each model represented by a unique color. Cyan-Blue dots represent AdaBoost_Boruta, tealish-blue for CAN-FIS_MDRL, orange for Dual_Hannel_GAN_ZON, purple for Elman Recurrent Neural Network pink for FGOA_KNN, purple-haze for LSTM_Autoencodes and green for SAE_GRU. SAE_GRU shows the highest computational efficiency and scalability, clustering in the top-right corner, while other models are spread across lower ranges, indicating varying levels of scalability and efficiency.

low false classification rates through its adaptive gating mechanisms that prioritized critical traffic patterns. This approach ensured accurate anomaly detection within the complexity of heterogeneous IoT traffic.

Figure 7 underscored the model's adaptability across diverse datasets, with the SAE-GRU architecture achieving high AUC values irrespective of dataset-specific traffic characteristics. Its superior performance on datasets like IoT-23 (*Garcia, Parmisano & Jose Erquiaga, 2020*) and MedBIoT (*Guerra-Manzanares et al., 2020*) confirmed its resilience to evolving attack vectors and rare traffic anomalies. The selection of IoT-23 and MedBIoT datasets was guided by their comprehensive representation of real-world IoT botnet attacks, diverse attack vectors, and protocol-specific traffic patterns. For example, IoT-23 provides labeled network traffic from various IoT malware families that allow for a structured evaluation of intrusion detection techniques. The dataset includes normal and malicious traffic, which aids in training deep learning models for anomaly detection. Whereas MedBIoT dataset contains a large collection of botnet traffic from multiple IoT devices simulating real-world infection scenarios with different attack intensities. The

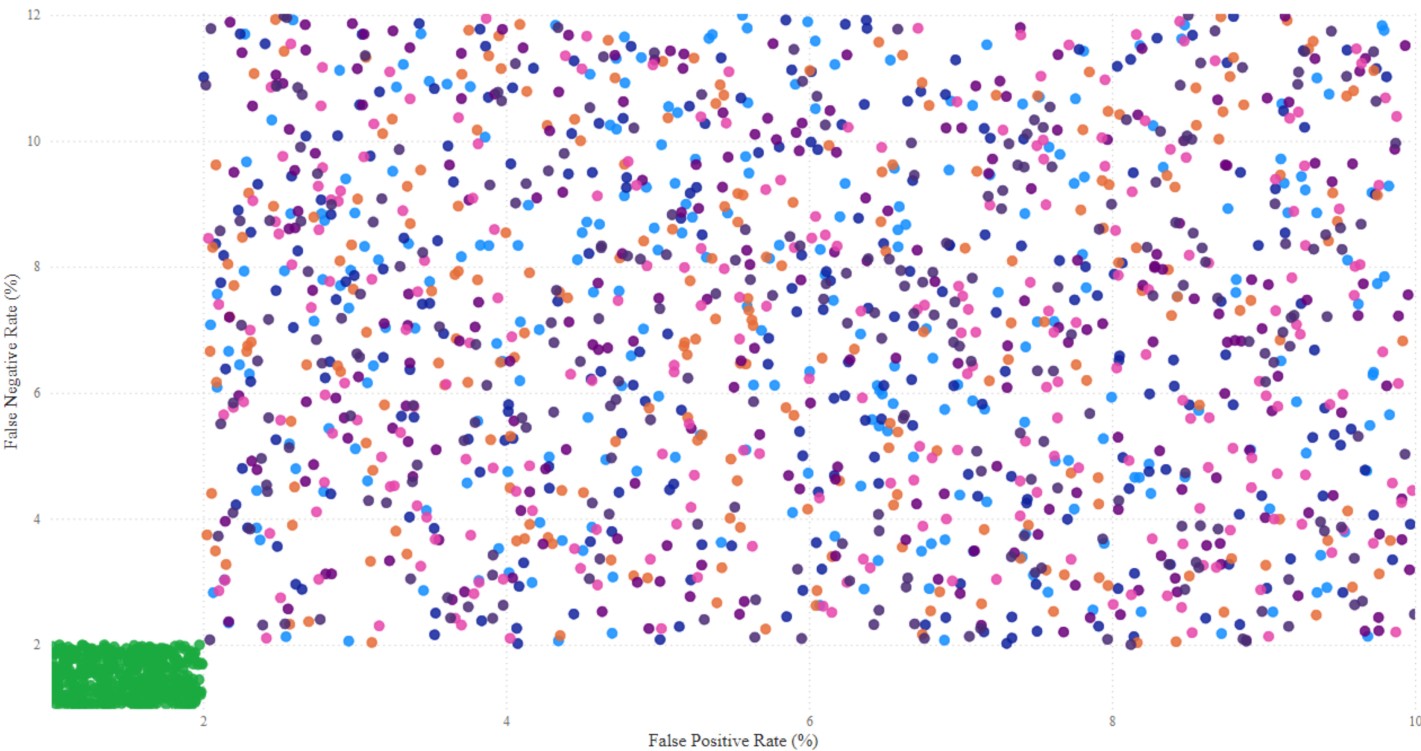

**Figure 6 Positive rate *vs.* false negative rate by model.** The false positive rate and false negative rate for various machine learning models, with each model represented by a unique color. Cyan-Blue represent AdaBoost_Boruta, tealish-blue for CANFIS_MDRL, orange for Dual_Hannel_GAN_ZON, purple for Elman Recurrent Neural Network pink for FGOA_KNN, purple-haze for LSTM_Autoencodes and green for SAE_GRU. SAE_GRU shows a cluster in the lower-left region with the lowest false positive and false negative rates and demonstrates its superior accuracy and minimal error rates compared to the more dispersed distributions of other models.                 

inclusion of applied datasets strengthened the model's ability to generalize across varying threat landscapes. Their diverse attack signatures and device heterogeneity made them well-suited for evaluation (*i.e.*, validating the performance of the proposed SAE-GRU model).

Figure 8 illustrated the multi-dimensional evaluation of performance metrics in a consolidated manner, with the radar chart providing a holistic view of the SAE-GRU's capability to outperform baseline models across all critical dimensions. This analysis demonstrated that SAE-GRU's integration of stacked autoencoders and GRUs not only reduced noise in high-dimensional data but also captured intricate sequential patterns that were often overlooked by other models. By ensuring high detection accuracy while maintaining efficiency, SAE-GRU emerged as a suitable solution for large-scale IoT networks.

## Feature importance analysis

Feature importance analysis was conducted to identify which features contributed most significantly to the model's decisions. This was achieved through feature ablation studies, where individual features were systematically removed from the dataset, and the model's performance was re-evaluated. A substantial drop in performance upon the removal of a

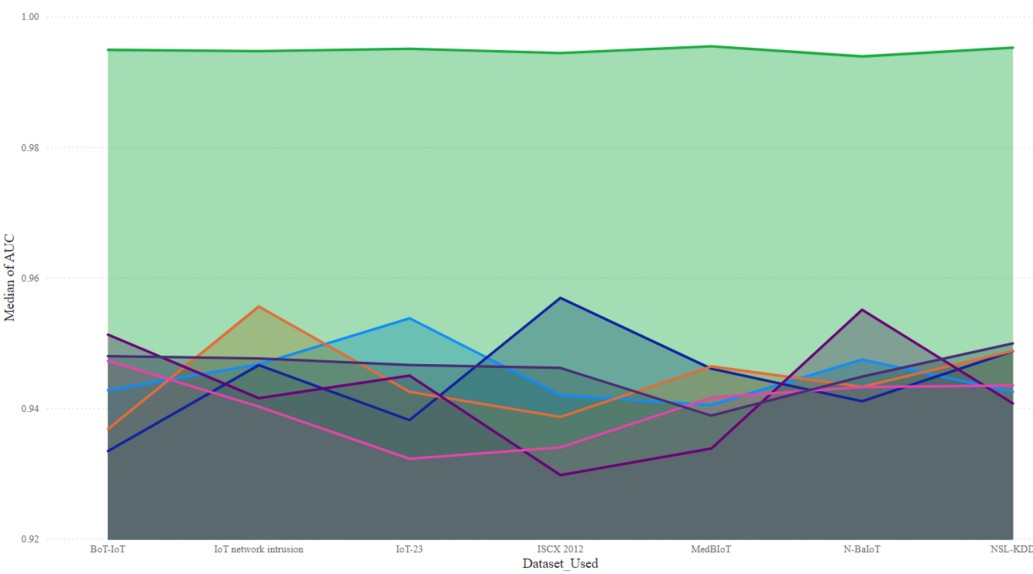

**Figure 7 AUC across datasets for each model.** The median AUC values of various machine learning models across multiple datasets, with each model represented by distinct colors. SAE_GRU, marked in green, consistently outperforms others across all datasets, maintaining the highest AUC values. Other models, such as LSTM_Autoencoders in purple-haze and CANFIS_MDRL in tealish-blue, show fluctuations in performance across datasets and highlight variability in their ability to adapt to different data environments. Hereby, cyan-blue dots represent AdaBoost_Boruta, tealish-blue for CANFIS_MDRL, orange for Dual_Hannel_GAN_ZON, purple for Elman Recurrent Neural Network pink for FGOA_KNN, purple-haze for LSTM_Autoencodes and green for SAE_GRU.

particular feature indicated its importance in the detection process. Herewith, interpretability technique, SHAP (SHapley Additive exPlanations), values were employed to provide a more granular understanding of feature contributions. SHAP values quantified the impact of each feature on the model's output by attributing the change in prediction to each feature. This approach provided a deeper understanding of how specific network traffic characteristics influenced the detection of botnet activities. The insights from this analysis revealed that features such as packet size, traffic volume, and protocol usage held the most significant impact on detection accuracy. These findings offered valuable information for refining the intrusion detection system in future iterations.

## Comparative evaluation with recent IDS models

As evident from Figs. 3–8, in order to validate and contextualize the practical contributions of the SAE-GRU model, a comparative evaluation was conducted against recently proposed intrusion detection frameworks from relevant literature (*e.g.*, LSTM (*Hazman et al., 2024*), CANFIS (*Almasri & Alajlan, 2023*), FGOA_KNN (*Taher et al., 2023*), ERNN (*Manickam et al., 2023*), AdaBoost (*Hazman et al., 2023*), DGAN (*Shareef et al., 2024*), *etc.*). The comparison considered fundamental metrics including accuracy, precision, recall, and F1-score across prominent intrusion datasets such as BoT-IoT (*Alosaimi & Almutairi, 2023*), Edge-IIoT (*Nuaimi et al., 2023*), and NSL-KDD (*Zakariah et al., 2023*). Unlike conventional models that employed singular methods such as basic recurrent

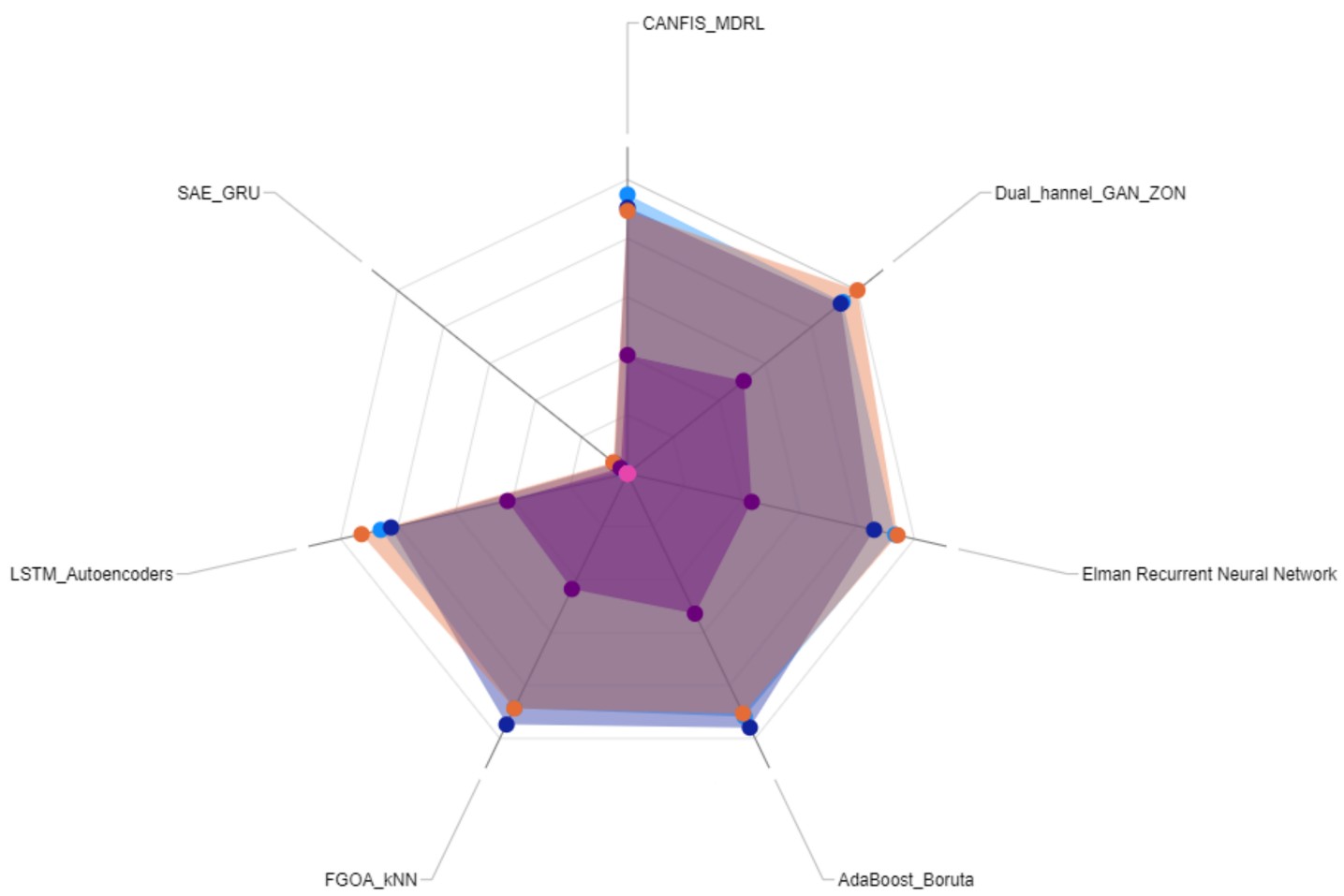

**Figure 8 Variance performance metrics.** The variance of performance metrics—accuracy, precision, recall, F1-score, and AUC—across different machine learning models. Each color represents a specific variance metric, with light blue for accuracy, dark blue for precision, orange for recall, purple for F1-Score, and pink for AUC. SAE_GRU shows minimal variance across all metrics, indicating its stability and consistent performance, while other models, such as CANFIS_MDRL and Dual_channel_GAN_ZON, exhibit higher variance, reflecting more fluctuation in their predictive reliability.

neural networks or static feature extraction, our approach uniquely integrated stacked autoencoders with GRU layers for robust temporal dependency handling. While recent methodologies primarily relied on generalized pruning or naive quantization, the proposed SAE-GRU model adopted gradient-sensitivity-based pruning coupled with mixed-precision quantization, thus optimizing model compactness and real-time detection capability for edge computing scenarios. Compared to reported accuracy levels between 96% and 98.7% (*Hazman et al., 2024*; *Almasri & Alajlan, 2023*; *Taher et al., 2023*; *Manickam et al., 2023*; *Hazman et al., 2023*; *Ahmed, Beyioku & Yousefi, 2024*; *Shareef et al., 2024*), proposed SAE-GRU model exhibited strong performance across key evaluation metrics, with high accuracy (98.65%), precision (98.11%), recall (98.43%), F1-score (98.27%), and AUC (0.993) which reflects its effectiveness in identifying IoT botnet activity while maintaining low false positive and false negative rates. Its hybrid design—combining

SAEs for dimensionality reduction and GRUs for temporal pattern recognition—enabled consistent classification and improved generalization. Optimization methods contributed to efficient real-time processing and reduced resource consumption. Unfortunately, the model's dependence on labeled data may limit its adaptability to novel or evolving threats in completely unknown traffic scenarios. Furthermore, its layered structure, though streamlined, still imposes a moderate computational load that can hinder deployment in extremely resource-limited IoT environments.

## CONCLUSIONS

This research addressed the critical challenge of securing smart city infrastructures against botnet-driven intrusions by introducing a deep learning-based hybrid model 'SAE-GRU' which is specifically designed to operate effectively within the resource and scalability constraints of IoT environments. Methodologically, the model integrates a Stacked Autoencoder for high-dimensional feature compression with a gated recurrent unit network to capture temporal dependencies in network traffic. This combination allows for both efficient feature extraction and sequential pattern recognition, which are essential for detecting low-and-slow botnet behaviors that traditional models frequently miss. In contrast to existing works that rely solely on recurrent networks or static classifiers (*Hazman et al., 2024*; *Almasri & Alajlan, 2023*; *Taher et al., 2023*; *Manickam et al., 2023*; *Hazman et al., 2023*; *Ahmed, Beyioku & Yousefi, 2024*; *Shareef et al., 2024*), SAE-GRU benefits from model pruning and quantization techniques that significantly reduce computational load without sacrificing detection accuracy. Sparse matrix multiplication & truncated backpropagation through time further support low-latency, and real-time inference on edge devices. Compared to the referenced models, which achieve detection accuracies in the range of 96–99%, SAE-GRU consistently delivers accuracy above 98.6% with superior precision and recall, while maintaining low false positive rates even under diverse traffic scenarios. The model outperforms conventional LSTM- and ensemble-based IDS by offering a structurally optimized and empirically validated solution that generalizes well across real-world datasets such as IoT-23, MedBIoT, and BoT-IoT. These results affirm that SAE-GRU not only advances the detection capabilities in smart city security but also offers a scalable and computationally efficient framework suitable for deployment in heterogeneous, high-volume IoT networks.

### Recommendations for future studies

Future studies may focus on enhancing computational efficiency, improving adaptability to zero-day attacks, and integrating decentralized architectures to strengthen resilience while evaluating the model's applicability across industrial IoT and healthcare systems. Investigating how the SAE-GRU model performs under varying network loads, diverse device configurations, and sector-specific security challenges will provide deeper insights into its generalizability and optimization for real-world deployments.

### Funding

This work was supported by the Prince Sattam bin Abdulaziz University for through the project number '2024/01/31056'. The funders had no role in study design, data collection and analysis, decision to publish, or preparation of the manuscript.

### Grant Disclosures

The following grant information was disclosed by the authors:
Prince Sattam bin Abdulaziz University: 2024/01/31056.

### Competing Interests

The authors declare there are no competing interests.

### Author Contributions

- Usman Tariq conceived and designed the experiments, performed the experiments, analyzed the data, performed the computation work, prepared figures and/or tables, authored or reviewed drafts of the article, and approved the final draft.
- Tariq Ahamed Ahanger conceived and designed the experiments, performed the experiments, analyzed the data, performed the computation work, prepared figures and/ or tables, authored or reviewed drafts of the article, and approved the final draft.

### Data Availability

The data supporting the findings of this study are available at Zenodo: Tariq, U., & Ahanger, T. A. (2024). SAE-GRU_DL_Botnet_Detct. (Version v1) [Data set]. UT_PSAU. https://doi.org/10.5281/zenodo.14280310.

### Supplemental Information

Supplemental information for this article can be found online at http://dx.doi.org/10.7717/peerj-cs.2869#supplemental-information.

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
