# Peer review of "Employing SAE-GRU deep learning for scalable botnet detection in smart city infrastructure"

_PeerJ Computer Science, doi:10.7717/peerj-cs.2869_

## Round 0.1 · original submission · Minor Revisions

Please consider the reviewer's suggestion. A contribution section is necessary to highlight the main importance, differences and other properties of the proposed idea. In addition, the comparison section should be extended by giving detailed discussion of current literature.

Reviewer 1 ·

Basic reporting

The main scientific question of this paper is as follows:

What solutions can be applied to botnet attack detection and prevention systems in IoT networks in the context of smart cities, considering high-dimensional data and temporal dependencies?

The paper addresses this question by proposing a hybrid deep learning model (SAE-GRU) that integrates Stacked Autoencoders (SAE) and Gated Recurrent Units (GRU) to address the issues related to data dimensionality and sequential analysis in heterogeneous IoT networks.

The paper presents a new deep learning model, SAE-GRU, designed to detect and mitigate botnet attacks in smart cities. The key elements are:
- Hybrid approach - SAE is used to reduce the dimensionality of the data, which improves computational efficiency. GRU analyzes sequential data, enabling the model to recognize temporal patterns related to botnet activity.
Test Environment—The model was evaluated in an emulated smart city environment, including various IoT devices and communication protocols.
- Experimental Results- The model achieved an average accuracy of 98.65% and high precision and recall values, indicating its effectiveness and scalability.
- Practical Significance- The proposed approach can be implemented in IoT monitoring systems to ensure the continuity of smart cities.

Strengths of the paper:
The combination of dimensionality reduction (SAE) with sequential analysis (GRU) significantly contributes to the development of intrusion detection systems in heterogeneous IoT environments.
The evaluation in an emulated smart city lends credibility to the results, demonstrating the proposed model's practical applicability.
- High accuracy, precision and recall suggest that the model effectively detects botnets, even in dynamic environments.
- Suggesting directions for further research, such as increasing computational efficiency and resistance to zero-day attacks, shows the potential for expanding the work.

Recommendations
- Added a section "Contributions- Highlight key contributions
- Shortened complex sentences and avoided overly technical language to increase the paper's readability.-

The paper presents a valuable contribution to developing IoT intrusion detection systems, especially in the context of smart cities. Once revised, the work will contribute solidly to IoT security research.

Experimental design

- Experimental Results- The model achieved an average accuracy of 98.65% and high precision and recall values, indicating its effectiveness and scalability.

Validity of the findings

Strengths of the paper:
The combination of dimensionality reduction (SAE) with sequential analysis (GRU) significantly contributes to the development of intrusion detection systems in heterogeneous IoT environments.
The evaluation in an emulated smart city lends credibility to the results, demonstrating the proposed model's practical applicability.
- High accuracy, precision and recall suggest that the model effectively detects botnets, even in dynamic environments.
- Suggesting directions for further research, such as increasing computational efficiency and resistance to zero-day attacks, shows the potential for expanding the work.

Additional comments

The main scientific question of this paper is as follows:

What solutions can be applied to botnet attack detection and prevention systems in IoT networks in the context of smart cities, considering high-dimensional data and temporal dependencies?

The paper addresses this question by proposing a hybrid deep learning model (SAE-GRU) that integrates Stacked Autoencoders (SAE) and Gated Recurrent Units (GRU) to address the issues related to data dimensionality and sequential analysis in heterogeneous IoT networks.

The paper presents a new deep learning model, SAE-GRU, designed to detect and mitigate botnet attacks in smart cities. The key elements are:
- Hybrid approach - SAE is used to reduce the dimensionality of the data, which improves computational efficiency. GRU analyzes sequential data, enabling the model to recognize temporal patterns related to botnet activity.
Test Environment—The model was evaluated in an emulated smart city environment, including various IoT devices and communication protocols.
- Experimental Results- The model achieved an average accuracy of 98.65% and high precision and recall values, indicating its effectiveness and scalability.
- Practical Significance- The proposed approach can be implemented in IoT monitoring systems to ensure the continuity of smart cities.

Strengths of the paper:
The combination of dimensionality reduction (SAE) with sequential analysis (GRU) significantly contributes to the development of intrusion detection systems in heterogeneous IoT environments.
The evaluation in an emulated smart city lends credibility to the results, demonstrating the proposed model's practical applicability.
- High accuracy, precision and recall suggest that the model effectively detects botnets, even in dynamic environments.
- Suggesting directions for further research, such as increasing computational efficiency and resistance to zero-day attacks, shows the potential for expanding the work.

Recommendations
- Added a section "Contributions- Highlight key contributions
- Shortened complex sentences and avoided overly technical language to increase the paper's readability.-

The paper presents a valuable contribution to developing IoT intrusion detection systems, especially in the context of smart cities. Once revised, the work will contribute solidly to IoT security research.

·

Basic reporting

The manuscript is well-structured and provides a clear narrative on the proposed SAE-GRU model for botnet detection in smart city infrastructures. However, there are areas where improvements can make the work more accessible and impactful:

Language and Clarity:
While the language is mostly clear, technical terms like "SAE-GRU" and "dimensionality reduction through stacked autoencoders" are used without sufficient explanation. Adding brief definitions would help readers unfamiliar with these concepts.
Some sentences, especially in the abstract and introduction, are quite long and could be simplified for better readability.

Introduction and Background:
The introduction does a good job of highlighting the importance of IoT security but would benefit from a concise summary of the specific problem being addressed and the knowledge gaps this research aims to fill.

References:
The paper references many relevant works but appears to miss a few key studies in IoT botnet detection. Including these would provide a more complete context.
Datasets like IoT-23 and MedBIoT are mentioned, but their relevance and suitability for this study need more explanation. A brief description of why these datasets were chosen would strengthen the argument.

Figures and Tables:
The figures are relevant and of good quality, but some, like Figure 4, lack detailed captions. Ensure all figures are labeled and fully explained in the text.
Table 5 is informative, but discussing its key insights in the main text would help readers better understand its significance.

Raw Data and Self-Containment:
The study mentions emulated data but does not provide details on how this data can be accessed. Sharing raw datasets and code repositories as supplementary materials would enhance transparency and reproducibility.
While the paper is self-contained, terms like "temporal dependencies" and "false positive rate" should be briefly defined to make the content clearer for a broader audience.

Experimental design

The experimental design is thorough and demonstrates a strong technical foundation, but a few areas could be improved to ensure clarity and reproducibility:

Research Question: While the study highlights the importance of IoT botnet detection, it would benefit from a more explicit statement of the specific knowledge gaps it aims to address. This would help readers better understand the significance of the work.

Methodology: The methodology is well-detailed, but some parts could be clearer. For example:
Data preprocessing steps, such as how missing values were handled or why specific normalization techniques were chosen, should be explained further.
Feature selection processes could be elaborated on to show how they contribute to the model's performance.

Dataset Selection: The paper makes good use of datasets like IoT-23 and MedBIoT, but more context on why these datasets were chosen and how representative they are of real-world IoT environments would strengthen the argument.

Reproducibility: To enable replication, it’s important to include details such as:
Hyperparameter settings for both the SAE and GRU components.
Hardware configurations and software libraries used.
Steps for replicating the emulated smart city environment.

Ethical Standards: It might also be helpful to briefly mention any ethical considerations related to the collection or simulation of data, especially when using IoT networks.

Overall, the experimental design is strong, but these additional details would enhance transparency and ensure the study is more accessible to a broader audience.

Validity of the findings

The findings of the study are well-presented and provide valuable insights into botnet detection in IoT networks. However, a few points need attention to ensure the validity and reliability of the conclusions:


Data and Robustness:
The emulated data used for testing is well-detailed, but the study would benefit from sharing the raw data and any preprocessing steps to allow for independent validation.
It’s not entirely clear how representative the chosen datasets (e.g., IoT-23, MedBIoT) are of real-world scenarios. Adding a discussion about their limitations and representativeness would strengthen the conclusions.

Statistical Soundness:
The study uses appropriate metrics like accuracy, precision, recall, and F1-score to evaluate performance. However, a detailed explanation of how the thresholds for these metrics were determined would provide more transparency.
The use of cross-validation is commendable, but the results could be enhanced by presenting confidence intervals or statistical tests to demonstrate the significance of the findings.

Conclusions and Link to Research Questions:
The conclusions are well-linked to the original research questions, but some claims, such as the scalability of the model, could benefit from more evidence or discussion about potential limitations.
It would be helpful to explicitly state how the proposed model addresses gaps in existing intrusion detection systems.

Generalizability:
The paper discusses the SAE-GRU model’s potential application in IoT networks, but its generalizability across diverse environments (e.g., industrial IoT or healthcare) needs more emphasis. Including a section on future work could address this.
By addressing these points, the findings will become even more robust and applicable to broader IoT security challenges.

Additional comments

The manuscript addresses a critical issue in IoT security, presenting a well-thought-out approach with the SAE-GRU model. The integration of dimensionality reduction and temporal pattern recognition is innovative and showcases strong technical expertise. However, to make the work even more impactful, consider the following:

Practical Implications: While the study demonstrates strong technical results, it would benefit from a clearer discussion of how this solution can be practically implemented in real-world smart city networks. Highlighting specific use cases or deployment scenarios could strengthen its relevance.

Comparisons with Existing Methods: The results are promising, but a more detailed comparison with other state-of-the-art methods (beyond basic metrics) would provide a clearer view of the model’s advantages and trade-offs.

Visual Representation: The use of figures and tables is effective, but a simplified summary or a flow diagram showing the end-to-end process (from data collection to anomaly detection) could help readers better understand the methodology.

Future Work: Including a section on potential extensions of this research—such as adapting the model for industrial IoT or enhancing scalability for larger datasets—would open up opportunities for further exploration.

Overall, this is a well-executed study that provides valuable contributions to the field of IoT security. With a few enhancements, it has the potential to be even more impactful.

---

## Round 0.2 · Major Revisions

Please make sure that all review comments have been considered. When I checked your manuscript, I found some weaknesses and problems. You should ensure that all required requests have been done properly. Here are some of them:

1-The organization of the manuscript should be reconsidered.

2-It should be detailed how the proposed model meets real-time processing needs and adequately manages the range of node-specific threats, which constitute the main motivation of the study.

3-Starting from the processing of data sets, the processes and flow followed in the proposed model should be technically summarized and visualized.

4-The information given between lines 119-187 provides background information on technical concepts rather than literature. Please update the flow.

5-Table 1 should also be supplemented with updated data sets.

6-Current literature has not been fully reviewed. The models proposed for the last five years in terms of smart city infrastructure should be considered comparatively by giving a comparison table in terms of properties, used techniques, and basic primitives. While examining literature studies, the number of studies examined should be increased, and the summary information presented should be made more compact.

7-The working concept of the proposed SAE-GRU model and how the features it contains are captured should be presented algorithmically, formally, or in a flow manner.

8-In model optimization, a technical explanation should be given on how and why pruning, weight quantization, and sparse matrix multiplication techniques are applied in the proposed model.

9-The layers, units, and components of the proposed model should be visualized and detailed relationally.

10-The manuscript needs proofreading.

11-The proposed model should be described with a general visual or flow diagram, and where and how the applied optimizations are applied should be detailed.

12-The lack of comparison with the literature leaves it unclear what contributions the presented model makes to the literature. The classifier method, used data sets, are of intrusion, accuracy, precision, and F1 score-like metrics need to be used in the comparison.

13-The analyses presented between lines 877-891 are meaningless. A separate comparison subheading should be created, and the proposed model should be compared in detail with the literature and conceptual and application results.

14-The results presented with the visuals in Figure 10-14 should be supported by detailed explanations.

15-A notation table is needed.

16-The most fundamental shortcomings of the study are ignoring the literature, explaining the working structure of the proposed model without technical details, and not explaining how the obtained features were not obtained technically.

**Language Note:** The Academic Editor has identified that the English language must be improved. PeerJ can provide language editing services - please contact us at [email protected] for pricing (be sure to provide your manuscript number and title). Alternatively, you should make your own arrangements to improve the language quality and provide details in your response letter. – PeerJ Staff

Reviewer 1 ·

Basic reporting

The authors responded to all my remarks. Now this article is ready for publication.

Experimental design

Everything is ok. The authors responded to all my remarks.

Validity of the findings

The authors responded to all my remarks.

Additional comments

I suggest publishing the article.

·

Basic reporting

The authors have improved clarity by shortening complex sentences and reducing overly technical language​
A "Significant Contributions" section was added to highlight key contributions​
The structure conforms to PeerJ standards, and references are well-cited.
Figures are appropriately labeled, but Figure 4 could still benefit from a more descriptive caption.

Remaining Suggestions:
Ensure all figures have fully descriptive captions to improve readability.

Experimental design

The research question is well-defined, and the study now explicitly states how it addresses IoT security challenges​
The authors have clarified dataset choices (IoT-23 and MedBIoT) but should further elaborate on how these datasets represent real-world IoT botnet activity.
Methods are now described in more detail, including model pruning, weight quantization, and k-fold cross-validation

Validity of the findings

The authors have added details on cross-validation and performance metrics, strengthening the study’s statistical rigor​
.
The conclusions are now more closely linked to the research questions.
Feature importance techniques like SHAP values have been added to validate detection accuracy

Additional comments

The revised manuscript is significantly improved, with a clearer focus and stronger justifications for methodology choices.

---

## Round 0.3 · Major Revisions

When creating a letter for those requested, for example, for comment 1, please provide guidance, such as the edits made to the article are given between lines 1-10. I want to follow the required changes in your manuscript by looking the lines. The problems are given as follows.

1. For your problem of exceeding the page limit, I suggest you increase the resolution of the images and tables to make them more compact. Then, you can scale them to make them much more compact. You can also write the results presented in Table 1 more concisely.
2. The studies you have reviewed in the Related Work section contain very extensive summaries. Summaries of approximately 25 lines complicate the scope of the study. Please present the reviews more briefly based on the specific primitives.
3. The Word document you uploaded as a clean version does not match the PDF file you edited. Please pay attention to this confusion and perform your next upload.
4. References 11-17 constitute your current literature. Please update Table 2 by examining the data sets used in the studies you selected as current literature as an additional column.
5. Addressing the information provided in the background Figures you presented with the old manuscript 1-7 verbally in the preliminaries section will create more comprehensive content in terms of explaining the components used in the proposed method. Please add background information about smart-city-related attack vectors, targets or impacts, botnet architecture, malware techniques, infection vectors, defense evasion techniques, etc., to the preliminaries section. Note that adding figures is not necessary.
6. Reading and trying to understand your proposed model section seems quite difficult. Understanding the basic components and the flow of the model is very difficult due to the complexity of the information given. Could you simplify them? In the proposed model section, first summarize the stages of your model. Maybe giving Figure 1 and then explaining each step in your model will be much clarified in terms of understanding your contribution. Present each component of the model in a simple and understandable way under subsections.
7. Add accessible resource code references of your Python and C++ implementation to line 712. Readers should be able to access and use your reference codes.
8. Can you reorganize the title of performance evaluation? After providing the technical information, the methods selected for comparison should be explained. Then, a comparative analysis of the proposed model should be presented with the Figures given in 3-8. Finally, adding comparison comments with the advantages and disadvantages of the proposed model in the examined metrics will make the comparison results more meaningful.
9. Can you strengthen the Conlusion section with technical explanations? Emphasizing what was used methodologically and what differences were obtained compared to the literature will express the importance of the proposed model.
10. Before Figures 3 to 8 are presented, explanations should be given as to why these methods were chosen for comparison.
11. The preliminary section should contain the definitions of the following techniques that have been claimed to be used.
• min-max normalization
• mutual information and variance thresholding techniques
• imputation techniques
• the feature scale normalization
• one-hot encoding
• weight decay
• L2 regularization
• Dropout and Early Stopping
• Pruning Optimization
• Weight quantization
• Sparse matrix multiplication

---

## Round 0.4 · accepted · Accept

All required reviewers' and editor's comments are done. So, the paper is ready for publication. Please check the final version and ensure there are no symbolic errors and grammatical errors in your paper.